# Performance-gated deliberation: A context-adapted strategy in which urgency is opportunity cost

**Maximilian Puelma Touzel**[1,2]\*, **Paul Cisek**[3], **Guillaume Lajoie**[1,4]

**1** Mila, Québec AI Institute, Montréal, Canada, **2** Department of Computer Science & Operations Research, Université de Montréal, Montréal, Canada, **3** Department of Neuroscience, Université de Montréal, Montréal, Canada, **4** Department of Mathematics & Statistics, Université de Montréal, Montréal, Canada

\* puelmatm@mila.quebec

**Data Availability Statement:** All relevant data are within the manuscript and its Supporting information files. All relevant simulation code are in a Github repository: https://github.com/mptouzel/dyn_opp_cost.

## Abstract

Finding the right amount of deliberation, between insufficient and excessive, is a hard decision making problem that depends on the value we place on our time. Average-reward, putatively encoded by tonic dopamine, serves in existing reinforcement learning theory as the opportunity cost of time, including deliberation time. Importantly, this cost can itself vary with the environmental context and is not trivial to estimate. Here, we propose how the opportunity cost of deliberation can be estimated adaptively on multiple timescales to account for non-stationary contextual factors. We use it in a simple decision-making heuristic based on average-reward reinforcement learning (AR-RL) that we call *Performance-Gated Deliberation* (PGD). We propose PGD as a strategy used by animals wherein deliberation cost is implemented directly as urgency, a previously characterized neural signal effectively controlling the speed of the decision-making process. We show PGD outperforms AR-RL solutions in explaining behaviour and urgency of non-human primates in a context-varying random walk prediction task and is consistent with relative performance and urgency in a context-varying random dot motion task. We make readily testable predictions for both neural activity and behaviour.

## Author summary

The value we place on our time impacts what we choose to do with it. Value our time too little, and we obsess over all details. Value it too much, and we rush carelessly to move on. How we value our time and how this value affects how much of it we allocate to tasks is not well-understood. The related cognitive processes are nevertheless thought to play a role in a wide range of diseases from Parkinson's to addiction. We propose a general strategy that balances the expected value of deliberation with the time spent, where time is valued according to recent performance. We found that recorded behaviour and brain activity from a previous experiment using non-human primates could be explained by this simple decision-making strategy. We show that this strategy explains how a brain signal called 'urgency', which limits how long subjects deliberate, varies with context. Our

**Funding:** MPT acknowledges support from IVADO via their postdoctoral fellowship award. PC acknowledges support from NSERC Discovery Grant (RGPIN-2016-05245). GL acknowledges support from FRQS Research Scholar Award, Junior 1 (LAJGU0401-253188), NSERC Discovery Grant (RGPIN-2018-04821), and the Canada CIFAR AI Chair program. The funders had no role in study design, data collection, and analysis, decision to publish, or preparation of the manuscript.

**Competing interests:** The authors have declared that no competing interests exist.

work helps to integrate the neuroscience of reward representations and the brain dynamics associated with deliberation.

## Introduction

Humans and other animals make a wide range of decisions throughout their daily lives. Any particular action usually arises out of a hierarchy of decisions involving a careful balance between resources. *Opportunity cost* [1] is the economics concept of the value of an alternative use that is lost when committing a limited resource to a given use. Opportunity costs are used to discount the value of allocating the resource to account for these forgone alternatives. A resource that is always limited is time. When to stop exploiting a local patch and leave to find another is central to time-limited patch foraging [2]. More generally, how long to deliberate in sequential decision-making settings is central to a wide range of tasks. The cost of *spending* time depends on its value, a construct that relies on comparing against the alternative things an agent could potentially do with it. Estimating time's value is not straightforward for a number of reasons. There are alternative choices at multiple decision levels, e.g. moving on from a job and moving on from a career, and each level requires its own evaluation. Moreover, the value of alternatives needs to be tracked as they may change over time depending on the context in which a decision is made. For example, animals will learn to value a given food resource differently depending on whether it is encountered during times of plenty versus scarcity [3]. The agent's knowledge of and ability to track context thus influences the value it assigns to possible alternatives. Factors that influence this knowledge have been shown to have direct consequences on behaviour.

These are significant, practical complications of making decisions contingent on opportunity cost. The opportunity cost of time is nevertheless well-studied in decision-making theory. It plays the role of a reference reward in definitions of relative value, most notably as the average reward in average-reward reinforcement learning (AR-RL) [4].

In neuroscience, AR-RL was first proposed to extend the reward prediction error hypothesis for phasic dopamine to account also for the observed properties of tonic dopamine levels [5]. It has since been used to emphasize the relative nature of reward-based decision-making [6] in explanations of human and animal behaviour in foraging [2], free-operant conditioning [7], perceptual decision-making [8, 9], cognitive effort/control [9, 10], and even economic exchange [11]. This perspective has been applied in clinical work through dopamine's relationship with vigor impairments. In particular, people's willingness to leave depleted patches have been shown to be influenced by dopaminergic depletion and restoration in Parkinson's disease [12] and dopaminergic drugs in healthy participants [13]. More generally, it provides a normative explanation for why deliberation times are shorter in contexts with high average reward and longer in contexts with low average reward due to the opportunity cost of time [14].

Unlike the alternative discount-reward approach, AR-RL is a theoretically well-defined and numerically stable formulation for long horizon decision problems [15], such as with task environments in which there is no definite end (known as *continuing environments* [16]). Solutions to AR-RL problems maximize average reward, in contrast to traditional fixed accuracy criteria in perceptual decision-making tasks that focus on maximizing trial reward alone [17]. The solutions to AR-RL formulations of tasks of long sequence of trials are decision boundaries in the state space of a trial. Determining this decision boundary requires maximizing the relative value, defined using the opportunity cost of time. The resulting optimal decision

boundaries typically 'collapse' over a trial: they cut deliberation short, e.g. in tasks where trial difficulty is variable [8, 18]. Up to now, however, AR-RL and most of its applications have focused on fixed contexts and have used the stationary average reward as the fixed opportunity cost of time, which ignores context-dependent performance variation. This is perhaps not surprising given that in psychological and neuroscientific studies of decision-making, we usually eliminate such contextual factors from the experimental design such that our models describe stationary behaviour. However, the brain mechanisms under study are adapted to a more diverse natural world in which changing environmental factors are often relevant, hard to infer and vary over time [6]. For example, the variance of rewards over contexts has long been shown to factor into preferences [19] and has made its way into modern studies on the range adaptation of reward representations [20].

We pursue a theory of approximate relative-value decision-making under uncertainty in a setting relevant to decision-making neuroscience. We first show that value in AR-RL can be expressed using the opportunity costs of deliberation and commitment. Here, the commitment cost is the shortfall in reward (relative to the maximum possible in a trial) that is expected to be lost when committing to a decision at a given time. The deliberation cost integrates the estimated cost rate of time. Highlighting the risk of value representations in non-stationary environments, we propose an approximation to the AR-RL value-optimal solution, Performance-Gated Deliberation (PGD). It uses the increasing opportunity cost of time in a trial to collapse the decision boundary directly, by-passing the need to maximize relative value. PGD thus reduces decision-making to estimating two opportunity costs: a commitment cost learned from the statistics of the environment and a deliberation cost estimated from tracking one's own performance in that environment. It explains how an agent, without explicitly tracking context parameters or storing a value function, can trade-off speed and accuracy according to performance at the typically longer timescales over which context changes. We propose that deliberation cost is then directly encoded as "urgency" in the neural dynamics underlying decision-making [8, 21–23]. The theory is thus directly testable using both behaviour and neural recordings.

To illustrate how PGD applies in a specific continuing decision-making task, and to make the links to a neural implementation explicit, we analyze behavior and neural recordings collected over eight years from two non-human primates (NHPs) [24, 25]. They performed successive trials of the "tokens task", a probabilistic guessing task in which information about the correct choice is continuously changing within each trial, and a task parameter controlling the incentive to decide early (the context) is varied over longer timescales. Behavior in the task, in both humans [22] and monkeys [25], provides additional support to an existing hypothesis about how neural dynamics implements time-sensitive decision-making [21]. Specifically, neural recordings in monkeys suggest that the evidence needed to make the decision predominates in dorsolateral prefrontal cortex [26]; a growing context-dependent urgency signal is provided by the basal ganglia [27]; and the two are combined to bias and time, respectively, a competition between potential actions that unfolds in dorsal premotor and primary motor cortex [24]. Similar findings have been reported in other tasks—for example, in the frontal eye fields during decisions about eye-movements [23]. We propose PGD as a theoretical explanation for why decision-making mechanisms are organized in this way. As an algorithm, it serves as a robust means to balance immediate rewards and the cost of time across multiple timescales. As a quantitative model, it serves to explain concurrently recorded behaviour and neural urgency in continuing decision-making tasks. From neural recordings in non-human primates and and behaviour in human and non-human primates, we show that it does so more accurately than AR-RL solutions. Adapting PGD to the random dot motion task in

which urgency was first characterized [23], we make quantitative predictions about neural urgency in such tasks, which we validate on their data within error bounds.

**Symbol glossary**. Highlighted in gray are parameters of the PGD model presented in this paper.

| symbol | quantity |
|---|---|
| $t$ | within-trial time |
| $k$ | trial index |
| $S_t$ | within-trial state at time $t$ |
| $\mathbf{S_t}$ | state sequence up to time $t$ |
| $R_k$ | reward of $k$th trial |
| $T_k$ | duration of $k$th trial |
| $t_k^{\mathrm{dec}}$ | decision time of $k$th trial |
| $\mathcal{C}_t^{\mathrm{del}}$ | within-trial opportunity cost of deliberation |
| $r_{\mathrm{max}}$ | maximum reward acheiveable in a trial |
| $b_t$ | belief of correct report given $\mathbf{S_t}$ |
| $\bar{r}_t$ | expected reward for reporting at time $t$ |
| $\mathcal{C}_t^{\mathrm{com}}$ | within-trial opportunity cost of commitment |
| $\rho$ | stationary reward rate |
| $\rho^*$ | optimal stationary reward rate |
| $\alpha$ | context parameter |
| $\rho_\alpha$ | context-conditioned stationary reward rate |
| $T_\alpha$ | context-conditioned stationary average trial duration |
| $\hat{\rho}_k^\tau$ | reward history filtered through a timescale, $\tau$ |
| $\tau_{\mathrm{long}}$ | a long timescale over which to estimate $\rho$ |
| $\tau_{\mathrm{context}}$ | a context-specific timescale over which to estimate $\rho_\alpha$ |
| $v$ | tracking cost sensitivity |
| $K$ | subjective reward scale factor |
| $T_{\mathrm{block}}$ | characteristic duration of a trial block |
| $c$ | auxiliary deliberation cost rate |
| $N_t$ | tokens difference |
| $p$ | jump probability of random walk, $p \geq 1/2$ |

# Results

## Theory of performance-gated deliberation

**Opportunity costs of deliberation and commitment, and drawbacks of average-reward reinforcement learning.**   We consider a class of tasks consisting of a long sequence of trials indexed by $k$ = 1, 2, ... (see Fig 1A), each of which provides the opportunity to obtain some reward by choosing correctly. In each trial, a finite sequence of states, $S_t$, $t$ = 0, ..., $t_{\mathrm{max}}$, is observed that provide evidence for an evolving belief about the correct choice among a fixed set of options. To keep notation simple, we suppress denoting the trial index, $k$, on quantities such as trial state, $S_t$, that also depend on trial time, $t$. The time of decision, $t_k^{\mathrm{dec}}$, and the chosen option determine both the reward received, $R_k$, and the trial duration, $T_k \geq t_k^{\mathrm{dec}}$. Importantly, decision timing can affect performance because earlier decisions typically lead to shorter trials (and thus more trials in a given time window), while later decisions lead to higher accuracy. Effectively balancing such speed-accuracy trade-offs is central to performing well in continuing episodic task settings. For a fixed strategy, the *stationary reward rate* (see slope of dashed

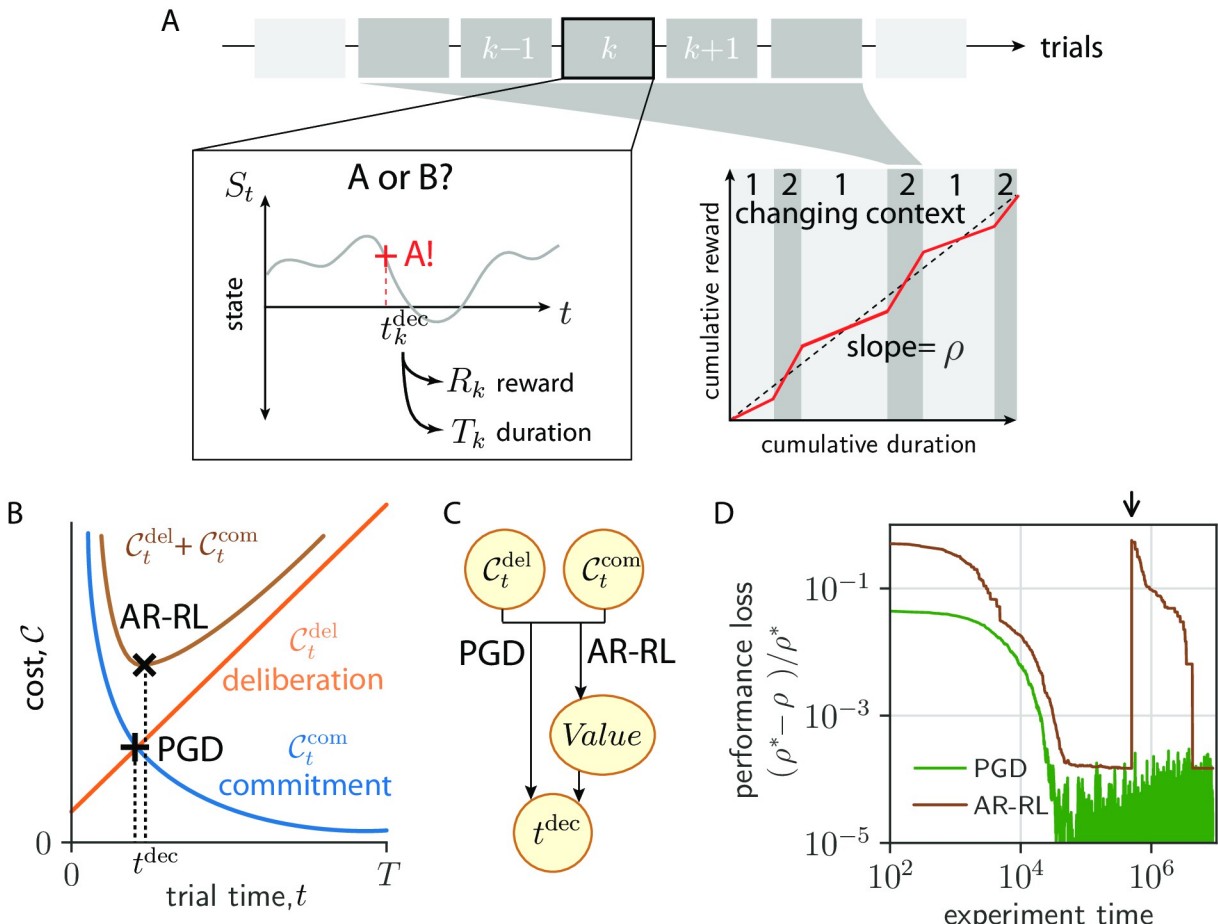

**Fig 1. AR-RL and performance-gated deliberation.** (A) Task setting. Left: Within trial state, $S_t$ evolves over trial time $t$ in successive trials indexed by $k$. The decision 'A' is reported at the decision time $t_k^{\text{dec}}$ (red cross), determining trial reward, $R_k$, and trial duration, $T_k$. Right: Sketch of cumulative reward versus cumulative duration. Context-conditioned reward rate (slope of red line), varies with alternating context (labelled 1 and 2) around average reward, $\rho$ (dashed line). (B) Decision rules based on opportunity costs of commitment, $\mathcal{C}_t^{\text{com}}$, and deliberation, $\mathcal{C}_t^{\text{del}}$. The AR-RL rule (black 'x') finds $t$ that minimizes $\mathcal{C}_t^{\text{del}} + \mathcal{C}_t^{\text{com}}$. The PGD rule (black cross) finds $t^{\text{dec}}$ at which they intersect, $\mathcal{C}_t^{\text{del}} = \mathcal{C}_t^{\text{com}}$. (C) Schematic diagram of each algorithm's dependency. PGD computes a decision time directly from the two opportunity costs, while AR-RL uses both to first estimate a value function, whose maximum specifies the decision time. (D) Loss (error in performance with respect to the optimal policy, $(\rho^* - \rho)/\rho^*$) over learning time in a patch-leaving task (AR-RL: brown, PGD: black). The arrow indicates when the state labels were randomly permuted.

line in Fig 1A(right)) is

$$\rho := \lim_{k \to \infty} \sum_k R_k \Big/ \sum_k T_k . \tag{1}$$

For a stochastic task environment, the definition of $\rho$ includes an average over different realizations of the task. Free-operant conditioning, foraging, and several perceptual decision-making tasks often fall into this class. Previous work [8, 28] has studied the belief of correct report for binary rewards, $b_t = P(R_k = 1 | S_t, t^{\text{dec}} = t)$, which also gives the expected trial reward, $\bar{r}_t = b_t \cdot 1 + (1 - b_t) \cdot 0 = b_t$ [8] (see [29] for more about the relationship between value-based and perceptual decisions). $S_t$ denotes the state sequence observed so far, $(S_0, \ldots, S_t)$. We consider greedy strategies that report the choice with the largest belief at decision time. The decision problem is then about *when* to decide.

Average-reward reinforcement learning (AR-RL), first proposed in artificial intelligence [30], was later incorporated into reward prediction error theories of dopamine signalling [5] and employed to account for the opportunity cost of time [7]. AR-RL was subsequently used to study reward-based decision-making in neuroscience and psychology [8, 9, 14, 31]. AR-RL centers around the average-adjusted future return, which penalizes the passage of time by using the average reward as a cost rate. Value is defined as this future return averaged over trial statistics. This average of a sum into the future of reward deviations from the average converges without a discount factor on account of the transient effects of conditioning the statistics on the state at which the decision is made. The AR-RL algorithms we consider aim to achieve the highest $\rho$ by also maximizing the average-adjusted value (see Methods for details). We now provide an alternative, but equivalent definition of average-adjusted trial return in terms of opportunity costs incurred by the agent.

We denote the opportunity cost of committing at time $t$ within a trial as $\mathcal{C}_t^{\text{com}}$, defined as the difference

$$\mathcal{C}_t^{\text{com}} = r_{\text{max}} - \bar{r}_t \,, \tag{2}$$

where $r_{\text{max}}$ is the maximum trial reward possible *a priori*. Within a trial, an agent lowers its commitment cost towards zero by accumulating more evidence, i.e. by waiting. Waiting, however, incurs another opportunity cost: the reward lost by not acting. We denote this opportunity cost of deliberation incurred up to a time $t$ in a trial as $\mathcal{C}_t^{\text{del}}$. In AR-RL, the constant opportunity cost rate of time is integrated so that for $T_k = t_k^{\text{dec}}$,

$$\mathcal{C}_t^{\text{del}} = \rho t \,. \tag{3}$$

With these definitions, the average-adjusted trial return for deciding at a time $t$ can be expressed as $r_{\text{max}} - (\mathcal{C}_t^{\text{com}} + \mathcal{C}_t^{\text{del}})$. It is maximized by jointly minimizing $\mathcal{C}_t^{\text{del}}$ and $\mathcal{C}_t^{\text{com}}$ (Fig 1B), giving the AR-RL optimal solution (see Methods for a formal statement and solution of the AR-RL problem). Expressed in this way, the average-adjusted trial return emphasizes the more general perspective that an agent's solution to the speed-accuracy trade-off is about how it balances the decaying opportunity cost of commitment and the growing opportunity cost of deliberation.

Despite their utility, value representations such as the average-adjusted trial return can be a liability in real world tasks where task statistics are non-stationary. To illustrate this, we consider the following foraging task. A foraging agent feeds among a fixed set of food (e.g. berry) patches. Total berries consumed in a patch saturates with duration $t$ according to a given saturation profile, shared across patches, as the fewer berries left are harder to find. Patches differ in their richness (e.g. berry density), which is randomly sampled and fixed over the task. Denoting patch identity (serving as context) by $s$, the food return is directly observed and deterministic given $s$. To perform well, the agent needs to decide when to move on from depleting the current patch. Further details about the task and its solution are given in the Methods. For a broad class of online AR-RL algorithms, the agent learns the average-adjusted trial return as a function of state and time. For a given patch, it then leaves when this return is at its maximum (*c.f.* Fig 1B). In Fig 1D, we show how the performance (brown line) approaches that of the optimal policy in time as the estimation of the AR-RL trial return improves with experience (see Methods for implementation details). However, if the agent's environment undergoes a significant disturbance (e.g. a forest fire due to which the patch locations are effectively re-sampled), the performance of this AR-RL algorithm can drop back to where it started. We implement such a disturbance via random permutation of the state labels at the time indicated by the arrow in Fig 1D. This is true over a range of learning rates and the

number of patches (S8 Fig). More generally, any approach that relies on estimating state-value associations shares this drawback, including those approaches that implicitly learn those associations by directly learning a policy instead [32]. Could context-dependent decision times be obtained without having to associate value or action to state? A means to do so is presented in the next section.

**Performance-gated deliberation.** We propose that instead of maximizing value as in AR-RL, which minimizes the sum of the two opportunity costs, $\mathcal{C}_t^{\mathrm{del}} + \mathcal{C}_t^{\mathrm{com}}$, the agent simply takes as its decision criterion when they intersect (shown as the black cross in Fig 1B).

$$t^{\mathrm{dec}} := \min_t\{t \mid \mathcal{C}_t^{\mathrm{del}} \geq \mathcal{C}_t^{\mathrm{com}}\} \quad \text{(PGD decision rule)} \quad (4)$$

We call this heuristic rule at the center of our results *Performance-Gated Deliberation* (PGD). Plotted alongside the AR-RL performance in Fig 1D for our example foraging task, PGD (black line) achieves better performance than AR-RL overall. It is also insensitive to the applied disturbance since PGD uses $\mathcal{C}_t^{\mathrm{del}}$ and $\mathcal{C}_t^{\mathrm{com}}$ directly when deciding, rather than as input to problem of optimizing average-adjusted value as in AR-RL (Fig 1C).

We constructed the above task so that PGD is the AR-RL optimal solution. In general, however, PGD is a well-motivated approximation to the optimal strategy, so we call it a heuristic. In the more general stochastic setting where there is residual uncertainty in trial reward at decision time, the PGD agent will have to learn the association between state and expected reward, $\bar{r}_t$. This association is learned from within-trial correlations only. In contrast, the opportunity cost of time as the basis for the deliberation cost depends on across-trial correlations that together determine the overall performance. It is thus more susceptible to non-stationarity. A typical task setting is when the value of the same low-level action plan differs across context. From hereon, we will assume the agent has learned the stationary opportunity cost of commitment and so focus on resolving the remaining problem: how to learn and use an opportunity cost of deliberation that exhibits non-stationarity on the longer timescales over which context varies.

**Reward filtering for a dynamic opportunity cost of deliberation.** The state disturbance in the toy example above altered task statistics at only a single time point. In general, however, changes in task statistics over time can occur throughout the task experience. A broader notion of deliberation cost beyond the static average reward is thus needed–one that can account for extended timescales over which performance varies. Such a cost serves as a dynamic reference in a relative definition of value based on a non-stationary opportunity cost of time. We first address how performance on various timescales can be estimated.

As a concrete example, we make use of the task that we will present in detail in the following section. This task has a context parameter, $\alpha$, that can vary in time on characteristic timescales longer than the moment-to-moment and can serve as a source of non-stationarity in performance. Here, the context sequence, $\boldsymbol{\alpha}_k$, varies on a single timescale, e.g. through periodic switching between two values. The resulting performance (Fig 2A(top)) varies around the stationary average, $\rho$ (purple), with context variation due to the switching (orange), as well as context-conditioned trial-to-trial variation (blue). The decomposition of time-varying performance into these multiple, timescale-specific components can be achieved by passing the reward signal through parallel filters, each designed to retain the signal variation specific to that timescale (Fig 2A(bottom)). There are multiple approaches to this decomposition. We chose a heuristic approach in which the performance over a finite memory timescale can be estimated by filtering the sequence of rewards through a simple low-pass filter [9, 33]. This filter is defined by an integration time, $\tau$, tuned to trade off the bias and variance of the estimate in order to best capture the variation on the desired timescale (e.g. how performance varies

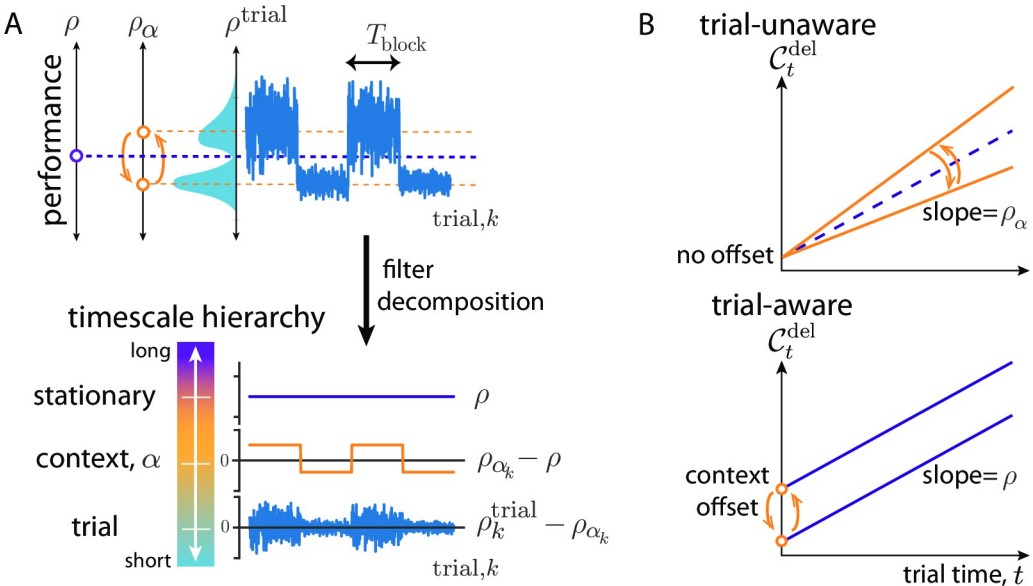

**Fig 2. Non-stationary opportunity cost.** (A) Top: Dynamics of trial performance ($\rho_k^{\text{trial}} \coloneqq R_k / T_k$; blue) with its distribution as well as dynamics of between context-conditioned averages of performance ($\rho_\alpha = \langle \rho_k^{\text{trial}} \rangle_{k|\alpha}$; orange), and the effectively stationary average performance ($\rho \sim \langle \rho_k^{\text{trial}} \rangle_k$; purple). Bottom: these are decomposed into a hierarchy by filtering reward history on trial, context, and long timescales, respectively. (B) Two hypothetical forms for context-specific trial opportunity cost. Top: Trial-unaware cost in which context varies the slope around $\rho$. Bottom: Trial-aware cost in which context variation is through a bias (*c.f.* Eq 5).

over different contexts). We denote such an estimate $\hat{\rho}_k^\tau$, and show in the Methods that it approximates the average reward over the last $\tau$ time units. We discuss the question of biological implementation in the discussion, but note here that the number and values of $\tau$ needed to represent performance variation in a given task could be learned or selected from a more complete set in an online fashion during task learning. In an experimental setting, these learned values can in principle be inferred from observed behaviour and we developed such an approach in the analysis of data that we present in the following section.

Applying this heuristic decomposition here, the stationary reward rate, $\rho$, can be estimated to high precision by using a long integration time, $\tau_{\text{long}}$, to the reward sequence $R_k$, producing the estimate $\hat{\rho}_k^{\tau_{\text{long}}}$. If $\boldsymbol{\alpha}_k$ were a constant sequence, $\mathcal{C}_t^{\text{del}} = \hat{\rho}_k^{\tau_{\text{long}}} t$, the stationary opportunity cost of deliberation Eq 3 of AR-RL. However, in this example context varies on a specific timescale, to which the former is insensitive. Thus, a second filtered estimate $\hat{\rho}_k^{\tau_{\text{context}}}$ is needed to estimate performance on this timescale. Unlike $\hat{\rho}_k^{\tau_{\text{long}}}$, this estimate tracks the effective instantaneous, context-specific performance, $\rho_{\alpha_k}$. Its estimation error arises from a trade-off, controlled by the integration time, $\tau_{\text{context}}$, between its speed of adaptation and its finite memory.

We consider two distinct hypotheses for how to extend AR-RL to settings where performance varies over context. The first hypothesis, $\mathcal{C}_t^{\text{del}} = \rho_\alpha t$, is the straightforward, *trial-unaware* extension of Eq 3, shown in Fig 2B(top). Here, performance is tracked only on a timescale sufficient to capture context variation and the corresponding cost estimate, $\hat{\rho}_{k-1}^{\tau_{\text{context}}}$, is incurred moment-to-moment, neglecting the trial-based task structure. However, this incorrectly lumps together two distinct opportunity costs: those incurred by moment-by-moment decisions and those incurred as a result of the effective planning implied by performance that varies over context. In particular, context is defined over trials not moments, and thus the

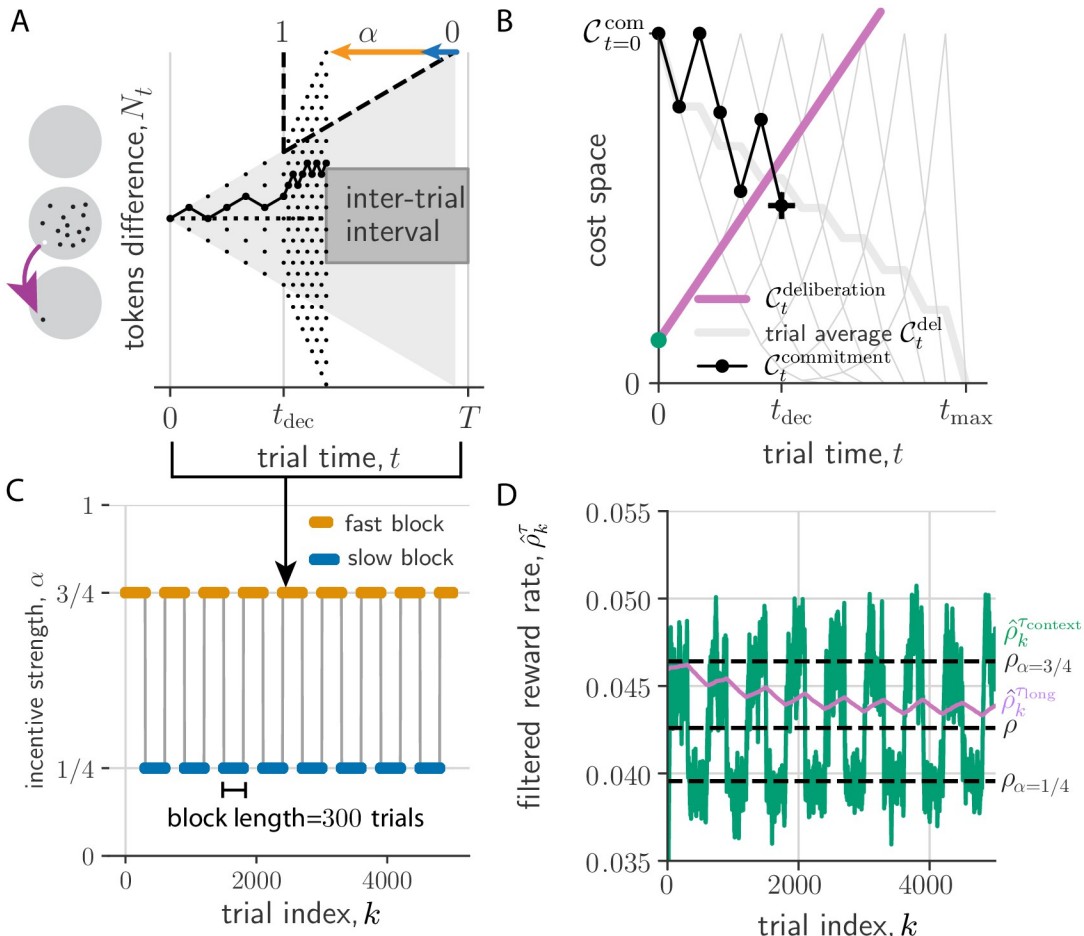

**Fig 3. PGD agent performs the tokens task for periodic context switching.** (A) A tokens task trial. Left: Tokens jump from a center to a peripheral region (gray circles). Right: The tokens difference, $N_t$, evolves as a random walk that accelerates according to $\alpha$ (here 3/4) post-decision time, $t^{\text{dec}}$. The trial duration is $T$, which includes an inter-trial interval. (B) Decision dynamics in cost space obtained from evidence dynamics in (A). Commitment cost trajectories (gray lattice; thick gray: trial-averaged) start at $\mathcal{C}^{\text{com}}_{t=0}$ and end at 0. Trajectory from (A) shown in black. $t^{\text{dec}}$ (black cross) is determined by the crossing of the commitment and deliberation cost. (C) Incentive strength switches between two values every 300 trials. (D) Expected rewards filtered on $\tau_{\text{long}}$ ($\hat{\rho}^{\tau_{\text{long}}}_k$, purple) and $\tau_{\text{context}}$ ($\hat{\rho}^{\tau_{\text{context}}}_k$, green). Black dashed lines from bottom to top are $\rho_{\alpha=1/4}$, $\rho$, and $\rho_{\alpha=3/4}$.

context-specific component of opportunity cost of a trial is a sunken cost paid at the outset of a trial. This inspires a second *trial-aware* hypothesis

$$\mathcal{C}^{\text{del}}_t = \rho t + (\rho_\alpha - \rho) T_\alpha . \quad \text{(trial-aware opportunity cost)} \tag{5}$$

Eq 5 is plotted over trial time $t$ in Fig 2B(bottom). Its first term is the AR-RL contribution from the stationary opportunity cost of moment-to-moment decisions using the stationary reward rate, $\rho$ estimated with $\hat{\rho}^{\tau_{\text{long}}}_k$. The second, novel term in Eq 5 is a context-specific trial cost deviation incurred at the beginning of each trial and computed as the average deviation in opportunity cost accumulated over a trial from that context ($T_\alpha$ is the average duration of a trial in context $\alpha$). This deviation fills the cost gap made by using the stationary reward rate $\rho$ in the moment-to-moment opportunity cost instead of the context-specific average reward, $\rho_\alpha$. This baseline cost derived from the orange time series in Fig 2A(bottom) vanishes in expectation, as verified through the mixed-context ensemble average reward (e.g. $\rho \equiv \Sigma_\alpha \rho_\alpha T_\alpha / \Sigma_\alpha T_\alpha$

when the context is distributed evenly among trials such that $\Sigma_\alpha(\rho_\alpha - \rho)T_\alpha = 0$). Thus, this opportunity cost reduces to that used in AR-RL when ignoring context, and suggests a generalization of average-adjusted value functions to account for non-stationary context. We estimate this baseline cost using $(\hat{\rho}_{k-1}^{\tau_{\text{context}}} - \hat{\rho}_{k-1}^{\tau_{\text{long}}})T_{k-1}$, where we have used the sample $T_{k-1}$ in lieu of the average $T_\alpha$. See S1 Fig for a signal filtering diagram that produces this estimate of Eq 5 from reward history. A main difference between the cost profiles from the two hypotheses is the cost at early times. Both the behaviour and neural recordings we analyze below seem to favor the second, trial-aware hypothesis Eq 5. We hereon employ that version in the main text, and show the results for the trial-unaware hypothesis in S7 Fig.

## Neuroscience application: PGD in the tokens task

In this section, we apply the PGD algorithm to the "tokens task" [22]. We first give a simulated example with periodic context dynamics. We then present an application to a set of non-human primate experiments in which context variation was non-stationary [25]. For the latter, we used the decision time dynamics over trials to fit a model for each of the two subjects. We then validated the models by assessing their ability to explain (1) the concurrently recorded behaviour via their context-specific behavioural strategies and (2) the neural activity in premotor cortex (PMd) via the temporal profile of the underlying neural urgency signals.

In the tokens task, the subject must guess as to which of two peripheral reaching targets will receive the majority of tokens that randomly jump, one by one every 200ms, from a central pool initialized with a fixed number of tokens. Importantly, after the subject reports, the time interval between remaining jumps contracts to once every 150ms (the "slow" condition) or once every 50ms (the "fast" condition), giving the subject the possibility to save time by taking an early guess. The interval contraction factor, $1 - \alpha$, for slow ($\alpha = 1/4$) and fast ($\alpha = 3/4$) condition is parametrized by $\alpha \in [0, 1]$, the incentive strength to decide early, which then serves as the task context. The subject is thus tasked with learning the statistics of the number of tokens at the end of the trial conditioned on an intermediate state in order to balance accuracy and deciding early.

In contrast to the patch leaving task example from Section A, the tokens task has many within-trial states and the state dynamics is stochastic. With the $t^{\text{th}}$ jump labelled $S_t \in \{-1, 1\}$ serving as the state, for the purposes of prediction, the history of states can be compressed into the tokens difference, $N_t = \sum_{i=1}^{t} S_i$, between the two peripheral targets with $N_0 = 0$. The dynamics of $N_t$ is an unbiased random walk (see Fig 3A), with its current value sufficient to determine the belief of a correct report, $b_t$ (computed in Methods). Since for binary rewards, $b_t$ is also the expected reward, $N_t$ is also sufficient for determining the opportunity cost of commitment, $C_t^{\text{com}}$ (Eq 2). We display this commitment cost dynamics in Fig 3B. It evolves on a lattice (gray), always starting at 0.5 (for $p = 1/2$) and ending at 0 for all $p$. We assume the agent has learned to track this commitment cost. The PGD agent uses this commitment cost, along with the estimate of the trial-aware deliberation cost, to determine when to stop deliberating and report its guess.

**A simulated example for a regularly alternating context sequence.**   We first show the behaviour of the PGD algorithm in the simple case where $\alpha$ switches back and forth every 300 trials (see Fig 3). We call such segments of constant $\alpha$ 'trial blocks', with context alternating between slow ($\alpha = 1/4$) and fast ($\alpha = 3/4$) blocks. The decision space in PGD is a space of opportunity costs, equivalent to the alternative decision space formulated using beliefs [8]. In particular, one can think of the deliberation cost as the decision boundary (Fig 3B). This boundary is dynamic (see S1 Video), depending on performance history via the estimates, $\hat{\rho}_k^{\tau_{\text{context}}}$ and $\hat{\rho}_k^{\tau_{\text{long}}}$, of the context-conditioned and stationary average reward, respectively. The

result of these dynamics is effective context planning: the PGD algorithm sacrifices accuracy to achieve shorter trial duration in trials of the fast block, achieving a higher context-conditioned reward rate compared to decisions in the slow block (*c.f.* the slopes shown in the inset of S2(D) Fig). This behaviour can be understood by analyzing the dynamics of $\hat{\rho}_k^{\tau_{\text{context}}}$ and $\hat{\rho}_k^{\tau_{\text{long}}}$, and their effect on the dynamics of the decision time ensemble.

The two performance estimates behave differently from one another solely because of their distinct integration times. Ideally, an agent would choose $\tau_{\text{context}}$ to be large enough that it serves to average over trial-to-trial fluctuations in a context, but short enough to not average over context fluctuations. In contrast, the value of $\tau_{\text{long}}$ would be chosen large enough to average over context fluctuations. We apply those choices in this simulated example, with rounded values chosen squarely in the range in which the values inferred from the behaviour in the following application will lie. As a result of this chosen values, the context estimate $\hat{\rho}_k^{\tau_{\text{context}}}$ relaxes relatively quickly after context switches to the context-conditioned stationary average performance (dashed lines in Fig 3D), but exhibits stronger fluctuations as a result. The estimate of the stationary reward, $\hat{\rho}_k^{\tau_{\text{long}}}$, on the other hand has relatively smaller variance. This variance results from the residual zigzag relaxation over the period of the limit cycle. Given the characteristic block duration, $T_{\text{block}}$, we can be more precise. In particular, when $T_{\text{block}}$ is much less than $\tau_{\text{long}}$ ($T_{\text{block}}/\tau_{\text{long}} \ll 1$), the within-block exponential relaxation is roughly linear. Thus, the average unsigned deviation between $\hat{\rho}_k^{\tau_{\text{long}}}$ and the actual stationary reward, $\rho$, can be approximated using $1 - \exp[-T_{\text{block}}/\tau_{\text{long}}] \approx T_{\text{block}}/\tau_{\text{long}} \ll 1$. This scaling fits the simulated data well (S2(D) Fig: inset).

The dynamics of these two performance estimates drives the dynamics of the *k*-conditioned decision time ensemble via how they together determine the deliberation cost (Eq 5; S1 Video). For example, the mean component of this ensemble relaxes after a context switch to the context-conditioned average, while the fluctuating component remains strong due to the sequence of random walk realizations (S2(C) Fig). In the case of periodic context, the performance estimates and thus also the decision time ensemble relax into a noisy periodic trajectory over the period of a pair of fast and slow blocks (Fig 3D). Over this period, they exhibit some stationary bias and variance relative to their corresponding stationary averages (distributions shown in S2(E) Fig).

**Fit to behavioural data from non-human primates and model validation.**　Next, we fit a PGD agent to each of the two non-human primates' behaviour in the tokens task experiments reported in [25] and compare to AR-RL solutions. As with the above example (*c.f.* Fig 3), trials were structured in alternating blocks of $\alpha = 1/4$ and $\alpha = 3/4$. Fig 4A shows context-switching $\alpha$-sequence from these experiments, which, in contrast to the above example exhibits large, irregular fluctuations in block size (These were primarily as as result of the experimenter adapting to fluctuations in motivation of the subject. D. Thura. Personal communication).

So far, PGD has only two free parameters: the two filtering time constants, $\tau_{\text{long}}$ and $\tau_{\text{context}}$. We anticipated only a weak dependence of the fit on the $\tau_{\text{long}}$, so long as it exceeded the average duration of a handful of trial blocks enabling a sufficiently precise estimate of $\rho$. In contrast, the context filtering timescale, $\tau_{\text{context}}$, is a crucial parameter as it dictates where the PGD agent lies on a bias-variance trade-off in estimating $\rho_{\alpha_k}$, the value of which determines the context-specific contribution to the deliberation cost (Eq 2). To facilitate the model's ability to fit individual differences, we introduce a subjective reward bias factor, *K*, that scales the rewards fed into the performance filters. We also add a tracking-cost sensitivity parameter, *v*, that controls $\tau_{\text{context}}$ to avoid wasting adaptation speed (see Methods for details). The latter made it possible to fit the asymmetric switching behaviour observed in the average decision time dynamics. With these four parameters, we quantitatively match the baselines and exponential-

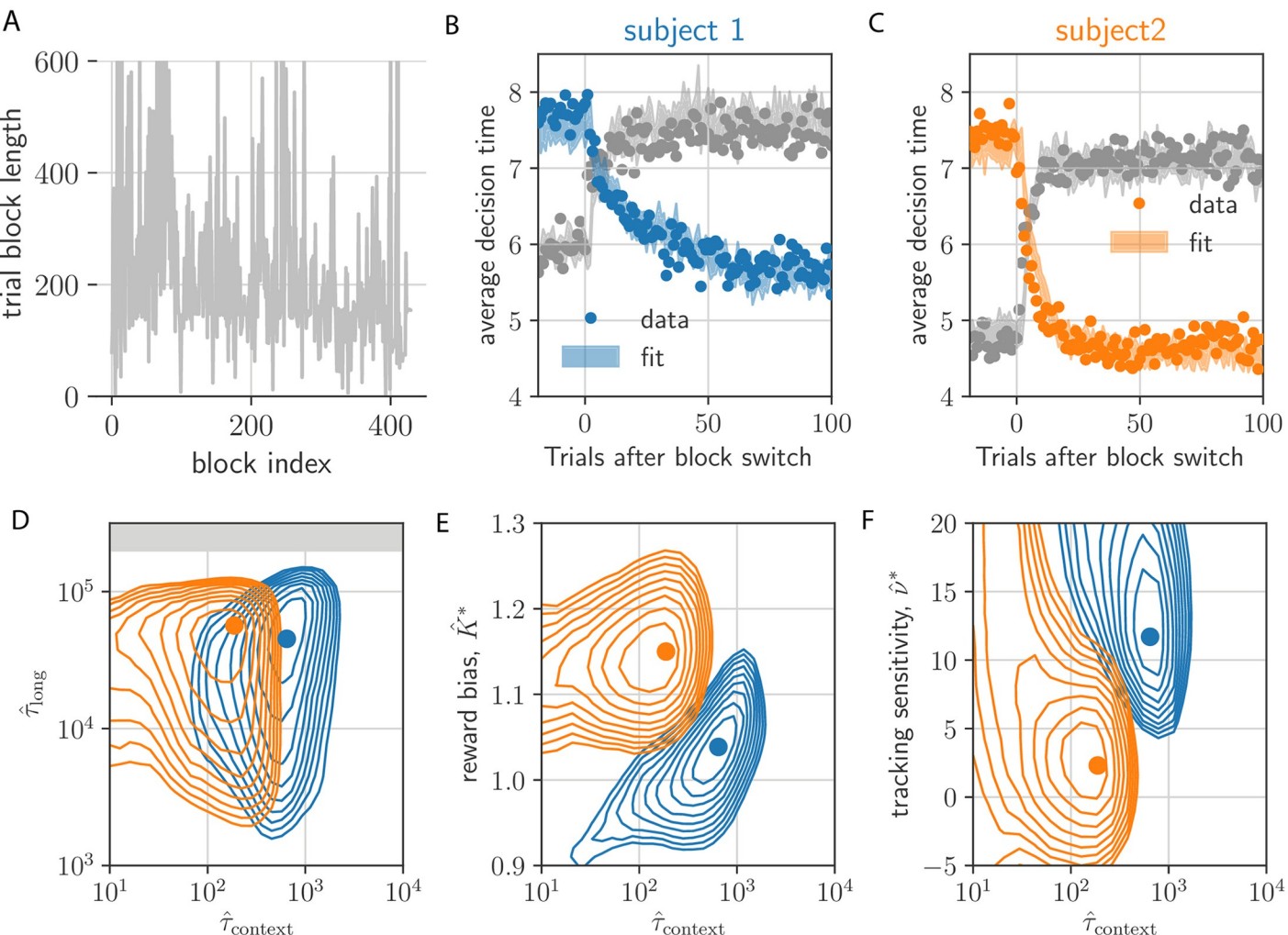

**Fig 4. PGD model fit to NHP behaviour for non-stationary $\alpha$-dynamics reported in Ref. [25].** (A) Block length sequence used in the experiment. (b,c) decision times (dots) aligned on the context-switching event type (fast-to-slow in gray; slow-to-fast in color) and averaged. Shaded regions are the standard error bounds of the models' average decision times. (D) Error evaluated on a $(\hat{\tau}_{context}, \hat{\tau}_{long})$-plane cut through the parameter space at the best-fitting $v = \hat{v}^*$ and $K = \hat{K}^*$ (gray area indicates timescales within an order of magnitude of the end of the experiment). Contours show the first 10 contours incrementing by 0.01 error from the minimum (shown as a circle marker). Colors refer to subject, as in (B) and (C). (E) Same for $(\hat{\tau}_{context}, \hat{K})$ at $\hat{\tau}_{long} = \hat{\tau}_{long}^*$ and $v = \hat{v}^*$. (F) Same for $(\hat{\tau}_{context}, \hat{v})$ at $\hat{\tau}_{long} = \hat{\tau}_{long}^*$ and $K = \hat{K}^*$.

like relaxation of the average decision time dynamics around the two context switches (Fig 4B and 4C; see Methods for fitting details).

A comparison of the best-fitting parameter values over the two monkeys (Fig 4D–4F) suggests that the larger the reward bias, $K$ (Fig 4E), the more hasty the context-conditioned performance estimate (the smaller $\tau_{context}$), and the lower the sensitivity to the tracking cost (Fig 4F). This is consistent with the hypothesis that subjects withhold cognitive effort in contexts of higher perceived reward [9]. Along with the correspondence in temporal statistics of the behaviour (e.g. S6 Fig), the fitted model parameters for the two subjects provides a basis on which to interpret the subject differences in the results of the next section, in particular their separation on a speed-accuracy trade-off, as originating in the distinct reward sensitivity shown here.

To better understand where both the data and the learned PGD agent lie in the space of strategies for the tokens task, we computed reward-rate (AR-RL) optimal solutions for a given

fixed context, $\alpha$ (here $\alpha \in [0, 1]$), using the same approach as [8] (we confirmed that the conventional discount-reward value iteration achieved the same solution in the limit of the undiscounted case). In each of average-reward and discount-reward formulations, the dynamic programming approach involves iterating Bellman's equation to obtain the optimal value functions from which the optimal policy and its reward rate can be obtained (see Methods for details). The optimal reward rate as a function $\alpha$ is shown in Fig 5A. The strategies generating these reward rates interpolate from the wait-for-certainty strategy at low $\alpha$ to the one-and-done strategy [34] at high $\alpha$. The former decides when the success probability first hits 1 and the latter decides after a single token jump in the direction of that jump. The $\alpha$-conditioned reward rates achieved by the two primates with their corresponding PGD model, and a reference human (Single subject behavioural data shared by Thomas Thierry.) are also shown in Fig 5A. They clearly fall below the optimal strategy, and, as expected, above the strategy that picks one of the three actions (report left, report right, and wait) at random.

To confirm that this similarity in performance between PGD and the data arises from a better fit to the behaviour than AR-RL, we plotted the distribution of the differences between model and data decision times, $|\Delta t_{\text{dec}}|$, conditioned on the context (Fig 5B and 5C). For comparison with previous work [8] and to account for deliberation cost in AR-RL, we added to the AR-RL reward objective a constant auxiliary deliberation cost rate, $c$, incurred up to the decision time in each trial, and chose the cost rate, $c^*$, that gave the lowest mean difference. In both contexts, PGD exhibits significantly lower error than this $c^*$ AR-RL solution (Kolmogorov-Smirnov two-sample test).

To reveal the source of this discrepancy in both performance and behaviour, we turned to analyzing the corresponding policies of PGD and $c$-based AR-RL agents. A robust and rich representation of the behavioural statistics is the state and time-conditioned survival probability that a decision has not yet occurred. It serves as a summary of the action policy associated with a stationary strategy (see Methods for its calculation from response times). Applied equally to the decision times of both model and data, it can provide a means of comparison even in this non-stationary setting. We give this conditional probability for each of the two contexts for subject 1 and its fitted PGD model in Fig 5D–5G. We left the many possible noise sources underlying the behaviour out of the model in order to more clearly demonstrate the PGD algorithm. However, such noise sources would be necessary to quantitatively match the variability in the data (e.g. added noise in the performance estimates leads to larger variability in the location of the decision boundary and thus also to larger spread in these survival probability functions). In the absence of these noise sources, we see the model underestimates the spread of probability over time and tokens state. Nevertheless, the remarkably smooth average strategy is well captured by the model (white dashed lines in Fig 5D–5G). Specifically, policies approximately decide once either of the peripheral targets receive a certain number of tokens. Comparing results across context, we find that fast block strategies (Fig 5E and 5G) exhibit earlier decision times relative to slow block strategies (Fig 5D and 5F) in both model and data. The strategies for subject 2 are qualitatively similar, but shifted to earlier times relative to subject 1 (S3 Fig). Our model explains this inter-individual difference as resulting from subject 2 having a larger reward bias, $K$, and faster context integration time, $\tau_{\text{context}}$ (c.f. Fig 4E).

The correspondence between the PGD model and data over the many token states in Fig 5D–5G explains their similar performance (c.f. Fig 5A). This similarity in policy is remarkable given that the model has essentially only a single, crucial degree of freedom ($\tau_{\text{context}}$), *a priori* unrelated to how decision times depend on token state. Note that in both the fitted PGD model and the primate behaviour, residual ambiguity ($N_t \approx 0$) is resolved at intermediate trial times (Fig 5B–5E).

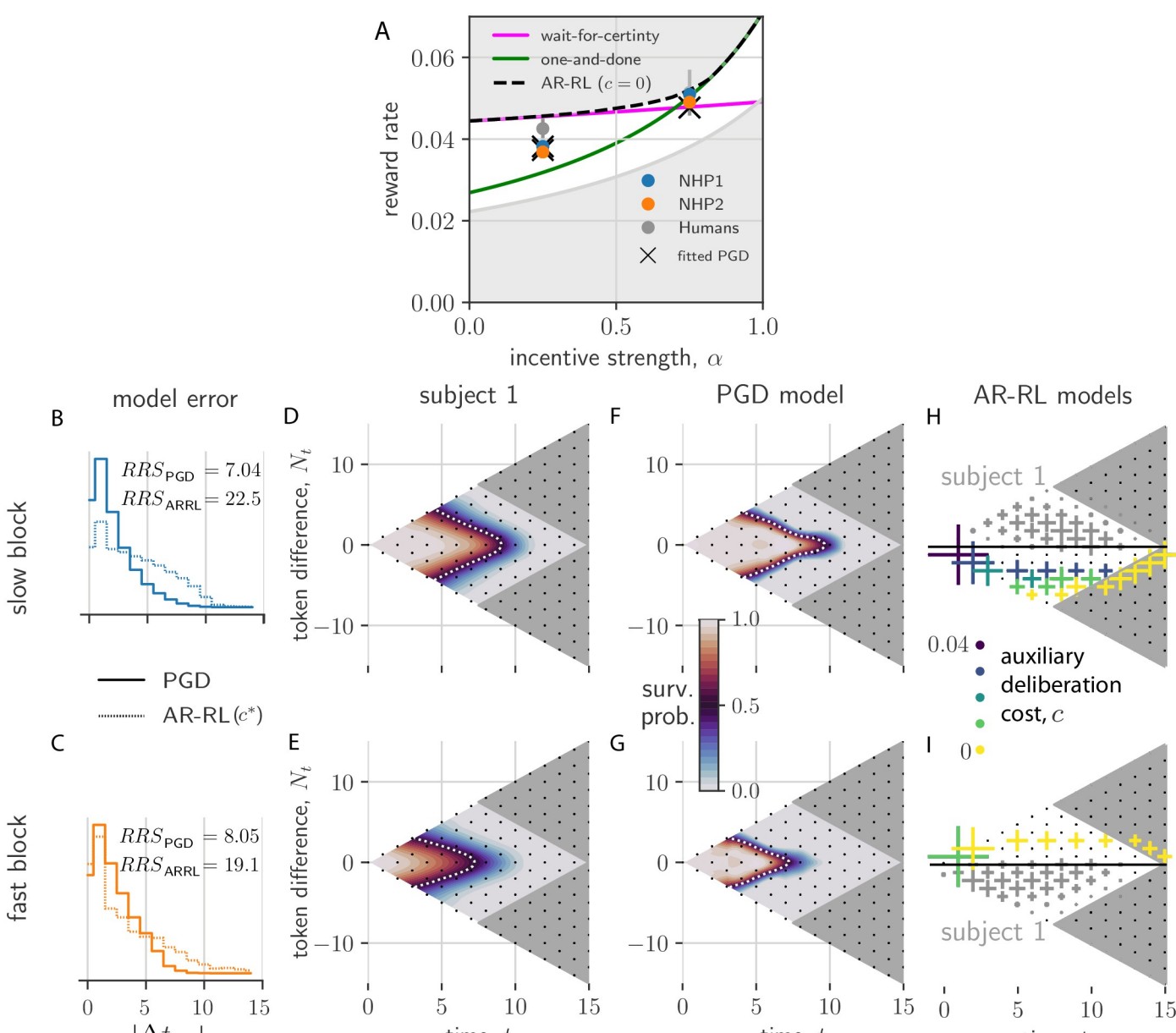

**Fig 5. Context-conditioned analysis of PGD and comparison to AR-RL models.** (A) Shown is the reward rate as a function of incentive strength, $\alpha$. The AR-RL solution with no augmented cost ($c = 0$) interpolates between the wait-for-certainty strategy (brown) and the one-and-done strategy (red). We also show the slow and fast context-conditioned reward rates for the two primates (blue and orange circles) and the PGD model fitted to them (crosses). For reference, we show the mean+/-std.dev. of a forthcoming dataset of 32 humans. Reward rates for the human and non-human primates are squarely in between the best (black dashed) and uniformly random (gray) strategy. (b,c) The distribution over trials of differences in decision times between model and data, $|\Delta t_{\text{dec}}| = |t_{\text{dec,data}} - t_{\text{dec,model}}|$, conditioned on slow and fast block contexts. Solid lines are for PGD. Dotted lines are for the AR-RL solution using the cost rate, $c^*$, with the lowest mean error. The residual sum of squares (RRS) for each model/block combination is displayed. (d-g) Interpolated state-conditioned survival probabilities, $P(t^{\text{dec}} = t|N_t, t)$, over slow (d,f) and fast (e,g) blocks. White dotted lines show the $P(t^{\text{dec}} = t|N_t, t) = 0.5$ contour. (h,i) State-conditioned decision time frequencies (cross size) from AR-RL optimal decision boundaries across different values of the cost rate, $c$ (colored crosses) for slow (h) and fast (i) conditions. Only samples with $N_t < 0$ and $N_t > 0$, respectively, are shown. For comparison, the reflected axes shows as gray crosses the state-conditioned decision time frequencies of the data.

The AR-RL strategies are plotted across $c$ in Fig 5G and 5H. In contrast, they give no intermediate decision times at ambiguous ($N_t \approx 0$) states, invariably waiting until the ambiguity resolves. This in fact holds over the entire ($\alpha$, $c$)-plane (see S9 Fig for the complete dependence), and also under the addition of a movement cost, i.e. a constant cost incurred by either of the reporting actions. Thus, whereas AR-RL policies shift around the edges of the relevant decision space as $\alpha$ or $c$ is varied, the PGD policy lies squarely in the bulk, tightly overlaying the policy extracted from the data. We conclude that the context-conditioned strategies of the non-human primates in this task are well-captured by PGD, while having little resemblance to the behaviour that would maximize reward rate with or without a fixed deliberation cost rate. We address the additional freedom of a time-varying cost rate in the discussion.

**Neural urgency and context-dependent opportunity cost.** So far, we have fit and analyzed the PGD model with respect to recorded behaviour. Here, we take a step in the important direction of confronting the above theory of behaviour with the neural dynamics that we propose drive it. The proposal for the tokens task mentioned at the end of the introduction has evidence strength and urgency combining in PMd, whose neural dynamics implements the decision process. In Fig 6A, we restate in a schematic diagram an implementation of this dynamics that includes a collapsing decision boundary. In the one-dimensional belief space for the choice (Fig 6A(top)) [8, 35], the rising belief collides with the collapsing boundary to determine the decision time. In the equivalent commitment and deliberation cost formulation developed here (Fig 6A(middle)), the falling commitment cost collides with the rising deliberation cost. The collapsing boundary in belief space can be parametrized as $C - u_t$, where $C$ is the initial strength of belief, e.g. some desired confidence, that is lowered by a growing function of trial time $u_t > 0$. The decision criterion is then $b_t > C - u_t$, where $b_t$ is the belief, i.e. the probability of a correct report. For AR-RL optimal policies, $u_t$ emerges from value maximization and thus has a complicated dependence on the opportunity cost sequence, $\mathcal{C}_t^{\mathrm{del}}$. For PGD, in contrast, $C$ is interpreted as the maximum reward $r_{\mathrm{max}}$ and $u_t$ is identically $\mathcal{C}_t^{\mathrm{del}}$. For a linear neural encoding model in which belief, rather than evidence, is encoded in neural activity, the

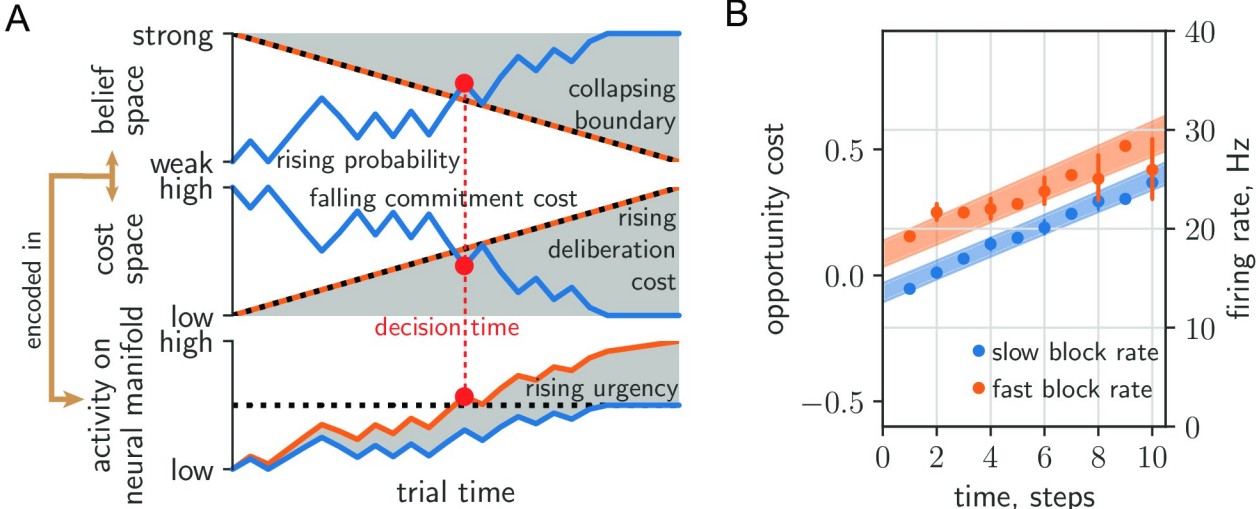

**Fig 6. Comparing neural urgency and collapsing decision boundaries.** (A) Top: Rising belief (blue) meets collapsing decision boundary (black dashed) in belief space. Middle: Falling commitment cost (blue) meets rising deliberation cost (black-dashed) in cost space. Bottom: Belief/commitment cost is encoded (blue) into a low-dimensional neural manifold, with the addition of an urgency signal (orange) (*c.f.* Figure 8 in [8]). The decision (red circle) is taken when the sum passes a fixed threshold (black-dashed). (B) Deliberation cost maps onto the urgency signal extracted from zero-evidence conditioned cell-averaged firing rate in PMd (200ms time steps).

sum of the encoded belief $\tilde{b}_t$ and the encoded collapsing boundary, $\tilde{u}_t$, evolve on a one-dimensional choice manifold. According to the proposal, when this sum becomes sufficiently large (e.g. $\tilde{b}_t + \tilde{u}_t > \tilde{C}$ for some threshold $\tilde{C}$), PMd begins to drive the activity in downstream motor areas towards the associated response.

Neural urgency was computed from the PMd recordings of [25] in [36]. This computation relies on the assumption that while a single neuron's contribution to $\tilde{b}_t$ will depend on its selectivity for choice (left or right report), the urgency $\tilde{u}_t$ is a signal arising from a population-level drive to all PMd neurons, irrespective of their selectivity. Thus, $\tilde{u}_t$ can be extracted from neural recordings by conditioning on zero-evidence states ($\tilde{b}_t = 0$) and averaging over cells. In [36], error bars were computed at odd times via bootstrapping; data at even times was obtained by interpolating between $N_t = \pm 1$; and data was pooled from both subjects. We have excluded times at which firing rate error bars exceed the range containing predictions from both blocks. To assess the correspondence of the components of the deliberation cost developed here and neural urgency, in Fig 6B we replot their result (*c.f.* Figure 8B of [36]). We overlay the mean (+/- standard deviation) of the opportunity cost sequence, $\mathcal{C}_t^{\mathrm{del}}$ (shaded area in Fig 4; averaged over all trials produced by applying the two fitted PGD models on the data sequence and conditioning the resulting average within-trial deliberation cost on context). To facilitate our qualitative comparison, we convert cost to spikes/step simply by adjusting the y-axis of the deliberation cost. The observed urgency signals then lie within the uncertainty of the context-conditioned deliberation cost signals computed from the fitted PGD models. There are multiple features of the qualitative correspondence exhibited in Fig 6B: (1) the linear rise in time; (2) the same slope across both fast and slow conditions; and (3) the baseline offset between conditions, where the fast condition is offset to higher values than the slow condition. Such features would remain descriptive in the absence of a theory. With the theory we have presented here, however, each has their respective explanations via the interpretation of urgency as the opportunity cost of deliberation: (1) the subject uses a constant cost per token jump, (2) this cost rate refers to moment-to-moment decisions, irrespective of context, that is reflective of the use of the context-agnostic stationary reward, and (3) trial-aware planning over contexts leads to an opportunity cost baseline offset with a sign given by the reward rate deviation $\rho_\alpha - \rho$ with respect to the stationary average, $\rho$.

Up to now, the computational and neural basis for urgency has remained largely unexplored in normative approaches, which also typically say little about adaptation effects (see [37] for a notable exception). In summary, we exploited the adaptation across context switches to learn the model and explained earlier responses in high reward rate contexts as the result of a higher opportunity cost of deliberation. While this qualitative effect is expected, we go beyond existing work by quantitatively predicting the average dependence on both time and state (Fig 5B–5E) as well as the qualitative form of urgency signal (Fig 6B). Taken together, the data is thus consistent with our interpretation that neural activity underlying context-conditioned decisions is gated by opportunity costs reflective of a trial-aware timescale hierarchy computed using performance estimation on multiple timescales.

## Discussion

We introduced PGD, a heuristic decision-making algorithm for continuing tasks that gates deliberation based on performance. We constructed a foraging example for which PGD is the optimal strategy with respect to the average-adjusted value function of average-reward reinforcement learning (AR-RL). While this will not be true in general, PGD does strike a balance between strategy complexity and return. The PGD decision rule does not depend on task

specifics and exploits the stationarity of the environment statistics while simultaneously hedging against longer term non-stationarity in reward context. It does so by splitting the problem into two fundamental components—learning the statistics of the environment in order to compute the opportunity cost of commitment, and tracking one's own performance in that environment with which to compute the opportunity cost of deliberation. This splitting is not only crucial to making efficient use of the opportunity cost of time in non-stationary settings. Building on the field's current understanding of how the cortico-basal ganglia system supports higher-level decision-making [38], we propose that the cost of deliberation arises from performance estimated on multiple, behaviourally-relevant timescales and is broadcast to multiple, lower-level decision-making areas to gate the speed of their respective evidence-driven attractor dynamics. Incorporating this cost into existing models of such dynamics [35, 39, 40] is an interesting direction for future work. Consistent with this picture, PGD's explanatory power was borne out at both the behavioural and neural levels for the tokens task data we analyzed. In particular, a deliberation cost constructed from trial-aware planning was supported independently by both these data sources. We used behavioural data to fit and validate the theory, and neural recordings to provide evidence of one of the neural correlates it proposes: the temporal profile of neural urgency.

## Scientific and clinical implications

In our proposal, we have linked two important and related, but often disconnected fields: the systems neuroscience of the neural dynamics of decision-making and the cognitive neuroscience of opportunity cost and reward sensitivity. The view that tonic dopamine encodes average reward is two decades old [5]. However, the existence of a reward representation decomposed by timescale has received increasing empirical support only in recent years, from cognitive results [41–43] to a recent unified view of how dopamine encodes reward prediction errors using multiple discount factors [44, 45] and of dopamine as encoding both value and uncertainty [46]. Dopamine's effect on time perception has been proposed [47] and has empirical support [48], but the mechanism by which its putative effect on decision speed is implicated in the neural dynamics of the decision-making areas driving motor responses was unknown. Our theory fills this explanatory gap by considering dynamic evidence tasks and parametrizing urgency using a multiple-timescale representation of performance. One candidate for the latter's neural implementation is in the complex spatio-temporal filtering of dopamine via release-driven tissue diffusion and integration via DR1 and DR2 binding kinetics [49]. Subsequent neural filtering and computation by striatal network activity could also play a role [50]. The study of spatiotemporal filtering of dopamine is increasingly accessible experimentally [51, 52] and provides an exciting direction for multiscale analysis of behaviour. Our proposal that urgency is the means by which the neural representation of reward ultimately affects neural dynamics in decision-making areas frames a timely research question on which these experimental methods could shed light.

We applied PGD to decisions playing out in PMd, a decision-making area relevant to arm movements. PGD appears to be relevant to other kinds of decisions, however. For instance, a large body of work has studied decisions through recordings in lateral intraparietal cortex in random dot motion tasks whose environment is formally similar to that of the tokens task. One seminal study identified an urgency signal with the same properties as those exhibited by the tokens task: a linear rise at early trial times that is independent of trial evidence and an offset with sign given by the reward rate deviation of the current context, here two and four-choice trials [23]. While decision boundaries obtained using AR-RL are evidence-independent, these models require tailored cost functions that are fit to those experiments in a procedure

that assumes optimality *a piori* [8]. Here, we offer an alternative explanation that behaviour is in fact suboptimal, with the decision boundary determsined directly by the estimated opportunity cost only. PGD decision boundaries are thus independent of evidence by construction. In contrast to the tokens task, however, context in these random dot task experiments was sampled randomly and thus its dynamics lacked temporal correlation [23]. In this case, a natural hypothesis from our approach is that a pair of performance filters, one for each context, tracks the reward history in two parallel streams. In this case, our theory would predict that the ratio of slopes of urgency across the two contexts reflects the ratio of context-conditioned reward rates. An estimation procedure described in the Methods for this data [23] agrees to within 20% error, providing support for the hypothesis that PGD underlies non-human primate behaviour on this widely-studied task. Within the context of the drift-diffusion models typically used to understand neural activity for that task, PGD provides a principled mechanism that implements collapsing decision boundary. PGD is thus easily incorporated into such models and testing the generality of our theory using tailored experiments in this setting is an important next step. We expect that the ideas behind PGD can also provide an explanation to the context dependence observed in response times in non-episodic tasks with varying context (e.g. in bandit tasks [53]) and foraging tasks.

Urgency may play a role in both decision and action processes, potentially providing a transdiagnostic indicator of a wide range of cognitive and motor impairments in Parkinson's disease and depression [54]. When fit to recorded behaviour from each subject across a set of subjects, our model can be used in clinical contexts to dissect inter-individual differences via the differences in the fitted model parameters. We gave an example here for the two non-human primates, explaining one as more hasty than the other due to higher reward bias and shorter memory. Our theory offers a means to ground these diverse results in neural dynamics by formulating opportunity cost estimation as the underlying causal factor linking vigor impairments (e.g. in Parkinson's disease) and dysregulated dopamine signalling in the reward system [54–56]. We provide a concrete proposal for a signal filtering system that extracts a context-sensitive opportunity cost from a reward prediction error sequence putatively encoded by dopamine. Neural recordings of basal ganglia provide a means to identify the neural substrate for this system.

### Commitment cost estimation

Beyond the estimation of the opportunity cost of deliberation, we assumed that the agent had a precise estimate of the expected reward, which it used to compute the within-trial commitment cost. For the tokens task, a recorded signal in dorsal lateral prefrontal cortex of non-human primates correlates strongly with belief [26], equivalent to the expected reward for binary rewards). How this quantity is computed by neural systems is not currently known. However, for a general class of tasks, a generic, neurally plausible means to learn the expected reward is via distributional value codes [46]. For example, the Laplace code is a distributional value representation that uses an ensemble of units over a range of temporal discount factors and reward sensitivities [57]. The authors show that expected reward is linearly decodeable from this representation.

### Experimental predictions

A feature of our decision-making theory is that it is highly vulnerable to falsification. First, with regards to behaviour via the shape of the action policy using our survival probability representation (*c.f.* Fig 5B–5E, 5G and 5H), PGD varies markedly with reward structure and thus provides a wealth of predictions for how observed behaviour should be altered by it. For

example, a salient feature of the standard tokens task is its reflection symmetry in the tokens difference, $N_t$. We can break this symmetry for which the theory predicts a distinctly asymmetric shape (S10 Fig; for details see Methods). Our theory is also prescriptive for neural activity via the temporal profile of neural urgency. The slope of $C_t^{\text{del}}$ remained fixed across blocks for relatively short block lengths used in the data analyzed here. In the opposite limit, $T_{\text{block}}/\tau_{\text{long}} \gg 1$, $\rho_k^{\tau_{\text{long}}}$ approaches $\rho_\alpha$ except when undergoing large, transient excursions after context switches. Thus, the deliberation cost is given by the first component in Eq 5 most of the time, with the context specific reward rate as the slope. One simple prediction is that the slope of urgency should exhibit increasing variation as the duration of the blocks increases.

### Reinforcement learning theory

We suggest how to generalize average-adjusted value functions to context-varying opportunity cost of time in a way that reduces to AR-RL when context is fixed or not tracked. This adds a continuing task perspective to episodic AR-RL, in line with recent work in machine learning, which is arguably the more appropriate reinforcement learning setting for many decision-making experiments in neuroscience. The epistemic perspective entailed in the estimation of these costs parallels a recent epistemic interpretation of the discount-reward formulation as encoding knowledge about the volatility of the environment [58].

Our work also suggests a new class of reinforcement learning algorithms between model-based and model-free: only parts of the algorithm need adjustment upon task structure variation. This is reminiscent of how the effects of complex state dynamics are decoupled from reward when using a successor representation [59], but tailored for the average-reward rather than the discount-reward formulation. We have left analysis of the algorithmic complexity of PGD to future work, but expect performance improvements, as with successor representations, in settings where decoupling the learning of environment statistics from the learning of reward structure is beneficial.

### Comparison with humans

In the space of strategies, PGD lies in a regime between fully exploiting assumed task knowledge (average-case optimal) and assumption-free adaptation (worst-case optimal). Highly incentivized human behaviour is likely to be more structured than PGD because of access to more sophisticated learning. While some humans land on the optimal one-and-done policy in the fast condition when playing the tokens task (Personal communication, Thomas Thierry), most do not. The human brain likely has all the components needed to implement PGD. Nevertheless, the situations in which we actually exploit PGD, if any, are as yet unclear. In particular, how PGD and AR-RL relate to existing behavioural models tailored to explain relative-value, context-dependent decision-making in humans [6], such as scale and shift adaptation [60], is an open question. Whether or not PGD is built into our decision-making, the question remains if PGD is optimal with respect to some bounded rational objective. In spite of the many issues with the latter approach [61], using it to further understand the computational advantages of PGD is an interesting direction for future work.

Despite our putative access to sophisticated computation, humans still exhibit measurable bias in how we incorporate past experience [62]. One simple example is the win-stay/lose-shift strategy, a more rudimentary kind of performance-gated decision-making than PGD, which explains how humans approach the rock-paper-scissors game [63]. In that work, numerical experiments demonstrated that this strategy outperforms at a population level the optimal Nash equilibrium for this game, demonstrating that the use of such seemingly sub-optimal strategies can confer a surprising evolutionary advantage. This example supports the claim

that relatively simple and nimble strategies such as PGD make for attractive candidates when acknowledging that a combination of knowledge and resource limitations over task, development, and evolutionary timescales have shaped decision-making in non-stationary environments.

## Methods

Code for simulations and main figure generation (written in Python 3) is publicly accessible as a online repository: https://github.com/mptouzel/dyn_opp_cost/.

### Patch leaving task

We devised a mathematically tractable patch leaving task for which PGD learning is optimal with respect to the average-adjusted value function. Here the value is simply the return from the patch. This value function is related, but not equivalent to the marginal value of optimal foraging, for which the decision rule is $C_t^{\text{del}} > r_{\max} - C_t^{\text{com}} = \bar{r}_t$ [2]). This choice of task allowed us to compare PGD's convergence properties relative to conventional AR-RL algorithms that make use of value functions. In contrast to PGD, the latter requires exploration. For a comparison generous to the AR-RL algorithm, we allowed it to circumvent exploration by estimating the value function from off-policy decisions obtained from the PGD algorithm using the same learning rate. We then compared them to PGD using their on-policy, patched-averaged reward. This made for a comparison based solely between the parameters of the respective models. If we did not allow for this, the AR-RL algorithms would have to find good learning signals by exploring. In any form, this exploration would lead them converge substantially slower. This setting thus provides a lower bound on the convergence times of the AR-RL algorithm.

In this task, the subject randomly samples (with replacement) $d$ patches, each of a distinct, fixed, and renewable richness defined by the maximum return conferred. These maximum returns are sampled before the task from a richness distribution, $p(r_{\max})$, with $r_{\max} > 0$ and are fixed throughout the experiment. The trials of the task are temporally extended periods during which the subject consumes the current patch. After a time $t$ in a patch, the return is defined $r(t) = r_{\max}(1 - (\lambda t)^{-1})$. This patch return profile, $1 - (\lambda t)^{-1}$, is shared across all patches and saturates in time with rate $\lambda$, a parameter of the environment that sets the reference timescale. The return diverges negatively for vanishing patch leaving times for mathematical convenience, but also evokes situations where leaving a patch soon after arriving is prohibitively costly (e.g. when transit times are long). A stationary policy is then a leaving time, $t_s$, for each of $d$ patches, where the $s$-subscript indexes the patch. Given any policy, the stationary reward rate for uniformly random sampling of patches is then defined as

$$\rho = \sum_{s=1}^{d} r_s(t_s) \bigg/ \sum_{s=1}^{d} t_s . \tag{6}$$

We designed this task to (1) emphasize the speed-return trade-off typical in many deliberation tasks, and (2) have a tractable solution with which to compare convergence properties of PGD and AR-RL value function learning algorithms.

A natural optimal policy is the one that maximizes the average-adjusted trial return, $Q(r, t) = r - \rho t$. Given the return profile we have chosen, the corresponding optimal decision time, $t_s^*$, in the $s$th patch obtained by maximizing $r - \rho t$ is $t_s^* = \sqrt{r_{\max,s}/(\lambda\rho)}$, which scales inversely with the reward rate so that decision times are earlier for larger reward rates, because consumption (or more generally deliberation) at larger reward rates costs more. We chose this

return profile such that stationary PGD learning gives exactly the same decision times: the condition $\mathcal{C}_t^{\text{del}} = \mathcal{C}_t^{\text{com}}$ for patch $s$ here takes the form $\rho t_s = r_{\max,s}/(\lambda t_s)$. Thus, they share the same optimal reward rate, $\rho^*$. Using $t_s^*$ for each patch in Eq 6 gives a self-consistency equation for $\rho$ with solution $\rho^* = \lambda \mu_1^2/4\mu_{1/2}^2$, where $\mu_n = \langle r_{\max}^n \rangle_{p(r_{\max})}$ (we have assumed $d$ is large here to remove dependence on $s$). Described so far in continuous time, the value function was implemented in discrete time such that the action space is a finite set of decision times selected using the greedy policy, $t^* = \operatorname{argmax}_t \hat{Q}(r, t)$, where $\hat{Q}(r, t)$ is the estimated trial return. As a result, there is a finite lower bound on the performance gap, i.e. the relative error, $\epsilon = (\rho^* - \rho)/\rho^* > 0$ for the AR-RL algorithm. Approaching this bound, convergence time for both PGD and AR-RL learning is limited by the integration time $\tau$ of the estimate $\hat{\rho}_k^\tau$ (*c.f.* Eq 8) of $\rho$. We note that PGD learns faster in all parameter combinations tested. To demonstrate the insensitivity of PGD to the state space representation, at $5 \times 10^5$ time steps into the experiment we shuffled the labels of the states. PGD is unaffected, while the value function-based AR-RL algorithm is forced to relearn and in fact does so slower than in the initial learning phase, due to the much larger distance between two random samples, than between the initial values (chosen near the mean) and the target sample.

## Filtering performance history

For unit steps of discrete time, the step-wise update of the performance estimate, $\hat{\rho}_t^\tau$, is

$$\hat{\rho}_t^\tau = (1 - \beta)\hat{\rho}_{t-1}^\tau + \beta R_t \, , \tag{7}$$

with $\beta = 1/(1 + \tau)$ called the learning rate, and $\tau$ the characteristic width of the exponential window of the corresponding continuous time filter over which the history is averaged. We add $\tau$ as a superscript when denoting the estimate to indicate this. Exceptionally, here $t$ indexes absolute time rather than trial time. Note that a continuous-time formulation of the update is possible via an event-based map given the decision times in which the reward event sequence is given as a sum of delta functions. In either case, to leading order in $\beta$, $\hat{\rho}_t^\tau \approx \beta \sum_i^t R_i$, i.e. the filter sums past rewards. Thus, when $\tau \sim \mathcal{O}(t) \gg 1$, $\beta \sim \mathcal{O}(1/t) \ll 1$ and so $\hat{\rho}_t^\tau \approx \beta \sum_i^t R_i \to \rho$ when $t$ is large.

The rewards in this task are sparse: $R_t = 0$ except when a trial ends and the binary trial reward $R_k$ (1 or 0) is received. A cumulative update of Eq 7 that smooths the reward uniformly over the trial duration and is applied once at the end of each trial is thus more compuatationally efficient. Resolving a geometric series leads to the cumulative update [9, 33]

$$\hat{\rho}_k^\tau = (1 - \beta)^{T_k}\hat{\rho}_{k-1}^\tau + (1 - (1 - \beta)^{T_k})\rho_k^{\text{trial}} \, , \tag{8}$$

where the smoothed reward, $\rho_k^{\text{trial}} = R_k/T_k$, can be interpreted as a trial-specific reward rate. The initial estimate, $\hat{\rho}_0^\tau$, is set to 0. Exceptionally, $\hat{\rho}_1^\tau = R_1/T_1$, after which Eq 8 is used. Using the first finite sample as the first finite estimate is both more natural and robust than having to adapt from zero. We will reuse this filter for different $\tau$ and denote the filtered estimate from its application with a $\tau$-superscript, $\hat{\rho}_k^\tau$. For example, the precision of $\hat{\rho}_k^{\tau_{\text{long}}}$ as an estimate of a stationary reward rate $\rho$ is set by how many samples it averages over, which is determined by the effective length of its memory given by $\tau_{\text{long}}$. Since we assume the subject has learned the expected reward, $\bar{r}_t$, we use it instead of $R_k$ when computing $\rho_k^{\text{trial}}$.

## Tokens task: A random walk formulation

The tokens task is a continuing task of episodes (here trials), which can be formulated using the token difference, $N_t$. Each trial effectively presents to the agent a realization of a finite-

length, unbiased random walk, $\boldsymbol{N}_{t_{\max}} = (N_0, \ldots, N_{t_{\max}})$ with $N_t = \{-t, \ldots, t\}$ and $N_0 = 0$. We express time in units of these steps. The agent observes the walk and reports its prediction of the sign of the final state, $\text{sign}(N_{t_{\max}}) = \pm 1$ ($t_{\max}$ is odd to exclude the case it has no sign). The time at which the agent reports is called the decision time, $t^{\text{dec}} \in \{0, 1, \ldots, t_{\max}\}$. For a greedy policy, $\text{sign}(N_t)$ can be used as the prediction (and the reporting action selected randomly if $N_{t^{\text{dec}}} = 0$). The decision-making task then only involves choosing when to decide. In this case, the subject receives reward $R = \Theta(N_{t_{\max}} N_{t^{\text{dec}}})$ at the end of the random walk, i.e. a unit reward for a correct prediction, otherwise nothing ($\Theta$ is the Heaviside function: $\Theta(x) = 1$ if $x > 0$, zero otherwise).

An explicit action space beyond decision time is not necessary for the case of greedy actions. It can nevertheless be specified for illustration in an Markov decision process (MDP) formulation: the agent waits ($a_t = 0$ for $t < t^{\text{dec}}$) until it reports its prediction, $a_{t^{\text{dec}}} = \pm$, after which actions are disabled and the prediction is stored in an auxiliary state variable used to determine the reward at the end of the trial. A MDP formulation for a general class of perceptual decision-making tasks, including the tokens and random dots task, is given in Methods).

Perfect accuracy in this task is possible if the agent reports at $t_{\max}$ since $R = \Theta(N_{t_{\max}}^2) = 1$. The task was designed to study reward rate maximizing policies. In particular, the task has additional structure that allows for controlling what this optimal policy is through the incentive to decide early, $\alpha$, incorporated into the trial duration for deciding at time $t$ in the trial,

$$T(t) = t + (1 - \alpha)(t_{\max} - t) + T_{\text{ITI}}. \tag{9}$$

Here, a dead time between episodes is added via the inter-trial interval, $T_{\text{ITI}}$, to make suboptimal the strategy of predicting randomly at the trial's beginning. It was set to 7.5 steps, the value used in [25]. In practise, this time also includes a random wait time (uniformly distributed between 2 and 3 steps) at the beginning of the trial to allow the subject to set-up. We excluded this time from our analysis. Only trials in which the subject completed the trial were analyzed. We emphasize that it is through the trial duration that $\alpha$ serves as a task parameter controlling the strength of the incentive to decide early. When $\alpha$ is fixed, we denote the corresponding optimal stationary reward rate, $\rho_\alpha$, obtained from the reward rate maximizing policy. This policy shifts from deciding late to deciding early as $\alpha$ is varied from 0 to 1 (c.f. S9(F) and S9(G) Fig).

We consider a version of the task where $\alpha$ is variable across two episode types, a slow ($\alpha = 1/4$) and fast ($\alpha = 3/4$) type. The agent is aware that the across-trial $\alpha$ dynamics are responsive (maybe even adversarial), whereas the within-trial random walk dynamics (controlled by the positive jump probability, here $p = 1/2$) can be assumed fixed (see the next section for how $p$ factors into the expression for the expected reward, $\bar{r}_t$.

## Expected trial reward for the tokens task

We derived and used an exact expression for the expected reward in a trial of the tokens task. We derive that expression here as well as a simple approximation. The state sequence is formulated as a $t_{\max}$-length sequence of random binary variables, $\boldsymbol{S}_{t_{\max}} = (S_1, \ldots, S_{t_{\max}})$, $S_t = \pm 1$, $i = 1$, $2, \ldots, t_{\max}$. Consider a simple case in which each is an independent and identically distributed Bernoulli sample, $P(s) = p^{\frac{1+s}{2}}(1 - p)^{\frac{1-s}{2}}$, for jump probability $p \geq 1/2$. The distribution of $\boldsymbol{S}_{t_{\max}}$ is then

$$P(\boldsymbol{s}_{t_{\max}}) = \prod_{i=1}^{t_{\max}} P(s_i). \tag{10}$$

We will use this distribution to compute expectations of quantities over this space of trajectories, namely the sign of $N_t = \sum_{i=1}^t S_i$, for some $0 \le t \le t_{\max}$ and in particular the sign of the final state, $\xi := \text{sgn}(N_{t_{\max}}) \in \{+, -\}$ given $N_t = n$. Note that $N_t$ is even if $t$ is even and same with odd values. We remove the case of no sign in $N_{t_{\max}}$ by choosing $t_{\max}$ to be odd.

First, consider predicting $\text{sgn}(N_t)$ with no prior information. The token difference, $-t \le N_t \le t$, appears directly in $P(\boldsymbol{s}_{t_{\max}})$. Marginalizing (here just integrating out) the additional degrees of freedom leads to a binomial distribution in the number of $S_i$ for $i \le t$ for which $S_i = +1$, $N_t^+ = \sum_{i=1}^t \Theta(s_i) = (t + N_t)/2$,

$$P(N_t^+ = n) = \binom{t}{n} p^n (1-p)^{t-n} , \tag{11}$$

with $n \in \{0, \ldots, t\}$ and $N_t = 2N_t^+ - t$. Thus, the probability that $N_t > 0$, i.e. $N_t^+ > t/2$, is

$$P(N_t > 0) = \sum_{n=0}^t \binom{t}{n} p^n (1-p)^{t-n} \Theta(n - t/2) . \tag{12}$$

Now consider predicting $\xi = \text{sgn}(N_{t_{\max}})$, given the observation $N_t = n$. Define $t' = t_{\max} - t$ as the remaining time steps to the predicted time and $N_{t'} = \sum_{i=t+1}^{t_{\max}} s_i$, i.e. the total count in the remaining part of the realization. Then the probability of $\xi = +$ conditioned on the state $N_t = n$, denoted $p_{n,t}$, is defined in the same way as $P(N_t > 0)$,

$$p_{n,t}^+ := P(\xi = +|N_t = n) = \sum_{n'=0}^{t'} \binom{t'}{n'} p^{n'} (1-p)^{t'-n'} \Theta(n' - (t' - n)/2) . \tag{13}$$

where $N_{t'}^+ = n'$ is the number of positive jumps in the remaining $t' = t_{\max} - t$ steps and we have used $N_{t_{\max}} = N_t + N_{t'} = N_{t'}^+ - (t' - N_t)/2$. The $\Theta(n' - (t' - n)/2)$ factor effectively changes the lower bound of the sum to $\max\{0, \lceil (t' - n)/2 \rceil\}$, where $\lceil \cdot \rceil$ rounds up. If $\lceil (t' - n)/2 \rceil \le 0$ then $p_{n,t}^+ = 1$ since the sum is over the domain of the distribution, which is normalized. Otherwise, the lower bound is $\lceil (t' - n)/2 \rceil$, and the probability of $\xi = +1$ is

$$p_{n,t}^+ = \sum_{n'=\lceil (t'-n)/2 \rceil}^{t'} \binom{t'}{n'} p^{n'} (1-p)^{t'-n'} . \tag{14}$$

For odd $t_{\max}$, the probability that $\xi = -$ is denoted $p_{n,t}^- = 1 - p_{n,t}^+$. For the symmetric case, $p = 1/2$,

$$p_{n,t}^+ = \frac{1}{2^{t'}} \sum_{n'=\lceil (t'-n)/2 \rceil}^{t'} \binom{t'}{n'} , \tag{15}$$

when $\lceil (t' - n)/2 \rceil > 0$ and 1 otherwise. This expression is equivalent to equation 5 in [22], which was instead expressed using $N_{t'}^-$.

The space of trajectories, i.e. of $\boldsymbol{s}_{t_{\max}}$, maps to a space of trajectories for $p_{n,t}^+$ defined on an evolving lattice in belief space. The expected reward in this case is,

$$\bar{r}_t := \langle r | N_t = n \rangle \quad = \mathbb{E}[\Theta(N_{t_{\max}} N_t) | N_t = n] \tag{16}$$

$$= \max\{p_{n,t}^+, 1 - p_{n,t}^+\} \tag{17}$$

$$= b_t , \tag{18}$$

where the belief of correct report $b_t := \max\{p_{n,t}^+, 1 - p_{n,t}^+\}$. The commitment cost $\mathcal{C}_t^{\text{com}} = r_{\max} - \bar{r}_t$, then also evolves on a lattice (see Fig 3B). More generally, $\bar{r}_t = \Delta r b_t + r_{\text{incorrect}}$ for $\Delta r$ the difference of correct $r_{\text{correct}}$ (here 1) and incorrect $r_{\text{incorrect}}$ (here 0) rewards. Since $r_{\max} = r_{\text{correct}}$, we have $\mathcal{C}_t^{\text{com}} = \Delta r (1 - b_t)$. For $p = 1/2$ and $\Delta r = 1$, $\mathcal{C}_{t=0}^{\text{com}} = 1/2$.

The shape of $p_{n,t}^+$ is roughly sigmoidal, admitting the approximation,

$$p_{n,t}^+ \approx \frac{1}{1 + \exp[-(at + b)n]} \tag{19}$$

where fitting constants $a$ and $b$ depend on $t_{\max}$. For $t_{\max} = 15$, $a = 0.03725$ and $b = 0.3557$. We demonstrate the quality of this approximation in S5 Fig. Approximation error is worse at $t$ near $t_{\max}$. More than 95% of decisions times in the data we analyzed occur before 12 time steps, where the approximation error in probability is less than 0.05. A similar approximation without time dependence was presented in [22]. We nevertheless used the exact expression Eq 15 in all calculations.

## PGD implementation and fitting to relaxation after context switches

We identified the times of the context switches in the data and their type (slow-to-fast and fast-to-slow). Taking a fixed number of trials before and after each event, we averaged the decision times over the events to create two sequences of average decision times around context switches (the result is shown in Fig 4A and 4B). We used a uniformly weighted squared-error objective, minimized with the standard (Nelder-Mead) simplex routine in python's scientific computing library's optimization package.

## Survival probabilities over the action policy

Behavioural analyses typically focus on response time distributions. From the perspective of reinforcement learning, this is insufficient to fully characterize the behaviour of an agent. Instead, the full behaviour is given by the action policy. In this setting, a natural representation of the policy is the probability to report as a function of both the decision time *and* the environmental state (see Fig 5). These are computed from the histograms of $(N_{t^{\text{dec}}}, t^{\text{dec}})$, over trials. However, the histograms themselves do not reflect the preference of the agent to decide at a particular state and time because they are biased by the different frequencies with which the set of trajectories visit each state and time combination. While there are obviously the same number of trajectories at early and late times, they distribute over many more states at later times and so each state at later times is visited less on average than states at earlier times. We can remove this bias by transforming the data ensemble to the ensemble of two random variables: the state conditioned on time $(N_t|t)$, and the event that $t = t^{\text{dec}}$. Conditioning this ensemble on the state gives $P(t = t^{\text{dec}}|N_t, t) = p(N_t, t = t^{\text{dec}}|t)/p(N_t|t)$. To reduce estimator variance, we focus on the corresponding survival function, $P(t < t^{\text{dec}}|N_t, t)$. So, $P(t < t^{\text{dec}}|N_t, t) = 1$ when $t = 0$ and decays to 0 as $t$ and $|N_t|$ increase. Unlike the unconditioned histograms, these survival probabilities vary much more smoothly over state and time. This justifies the use of the interpolated representations displayed in Fig 5B–5E. Note that to simplify the analysis, we have binned decision times by the 200 ms time step between token jumps. This is justified by the small deviations from uniformity of decision times modulo the time step shown in S11 Fig.

## Episodic decision-making and dynamic programming solutions of value iteration

We generalize the mathematical notation and description of an existing AR-RL formulation and dynamic programming solution of the random dots task [8], a binary perceptual evidence

accumulation task extensively studied in neuroscience. To align notation with convention in reinforcement learning theory, exceptionally here $s$ denotes the belief state variable, ie. a representation of the task state sufficient to make the decision (e.g. the tokens difference, $N_t$, in the case of the tokens task). We connect this extended formulation to account for a dynamic deliberation cost. We write it in discrete time, though the continuous time version is equally tractable.

The problem is defined by a recursive optimality equation for the value function $V(s|t)$ in which the highest of the action values, $Q(s, a|t)$, is selected. We formalize the non-stationarity within episodes by conditioning on the trial time, $t$, where $t = 0$ is the trial start time. $Q(s, a|t)$ is the action-value function of average-reward reinforcement learning [15], i.e. the expected sum of future reward deviations from the average when selecting action $a$ when in state $s$, at possible decision time $t$ within a trial, and then following a given action policy $\pi$ thereafter. The action set for these binary decision tasks consists of *report left* (−), *report right* (+), and *wait*. When *wait* is selected, time increments and beliefs are updated with new evidence. We use a decision-time conditioned, expected trial reward function, $r(s, a|t)$ with $a = \pm$, that denotes the reward expected to be received at the end of the trial after having reported ± in state $s$ at time $t$ during the trial. Note that $r(s, a|t)$ can be defined in terms of a conventional reward function $r(s, a)$ if the reported action, decision time, and current time are stored as an auxiliary state variable so they can be used to determine the non-zero reward entries at the end of the trial.

The average-reward formulation of $Q(s, a|t)$ naturally narrows the problem onto determining decisions within only a single episode of the task. To see this, we pull out the contribution of the current trial,

$$Q(s, a|t) \quad = \mathbb{E}^{\pi}\left[\sum_{t'=t}^{T} R_t - \rho \bigg| S_t = s, A_t = a\right] + V(s|T+1) \tag{20}$$

where $T$ is the (possibly stochastic) trial end time and $V(s|T+1)$ is the state value at the start of the following trial, which does not depend on $s_t$ and $a_t$ for independently sampled trials. Following conventional reinforcement learning notation, the expectation $\mathbb{E}^{\pi}$ is over all randomness conditioned on following the policy, $\pi$, which itself could be stochastic [15]. When trials are identically and independently sampled, the state at the trial start is the same for all trials and denoted $s_0$ with value $V_0$. Thus, the value at the start of the trial $V(s|t = 0) = V(s|T+1) = V_0$ equals that at the start of the next trial and so, by construction, the expected trial return (total trial rewards minus trial costs) must vanish (we will show this explicitly below). Note that the value shift invariance of E1 20 can be fixed so that $V_0 = 0$.

The *optimality equation* for $V(s|t)$ arises from a greedy action policy over $Q(s, a|t)$: it selects the action of the largest of $Q(s, -|t)$, $Q(s, +|t)$, and $Q(s, wait|t)$. The value expression for the wait-action is incremental, and so depends on the value at the next time step. In contrast, expression for the two reporting actions integrates over the remainder of the trial since no further decision is made and so depends on the value at the start of the following trial. The resulting optimality equation for the value function $V(s|t)$ is then

$$V(s|t) \quad = \max_{a} Q(s, a|t) \, ,$$

$$Q(s, \pm|t) \quad = r(s, \pm|t) - \sum_{t'=t+1}^{T} c_{t'} + V(s|t = T+1) \, , \tag{21}$$

$$Q(s, wait|t) \quad = -c_t + \mathbb{E}_{s_{t+1}|s}[V(s_{t+1}|t+1)] \, ,$$

$$V(s|t = 0) \quad = V(s|t = T+1) \, .$$

Here, $t = 0, 1, \ldots, t_{\max}$ within the current trial and $t = T + 1, T + 2\ldots$ in the following trial, with $t_{\max}$ the latest possible decision time in a trial, and $T = T(t)$ the decision-time dependent trial duration. For inter-trial interval $T_{\mathrm{ITI}}$, $T$ satisfies $T_{\mathrm{ITI}} \leq T \leq t_{\max} + T_{\mathrm{ITI}}$. $c_t$ is the cost rate at time $t$. The second term in $Q(s, \mathit{wait}|t)$ uses the notation $\mathbb{E}_{x|y}[z]$, i.e. the expectation of $z$ with respect to $p(x|y)$. The last line in Eq 21 is the self-consistency criterion imposed by the AR-RL formulation, which demands that the expected value at the beginning of the trial be the expected value at the beginning of the following trial. The greedy policy then gives a single decision time for each state trajectory as the first time when $Q(s, -|t) > Q(s, \mathit{wait}|t)$ or $Q(s, +|t) > Q(s, \mathit{wait}|t)$, with the reporting action determined by which of $Q(s, -|t)$ and $Q(s, +|t)$ is larger. For given $c_t$, dynamic programming provides a solution to Eq 21 [8] by recursively solving for $V(s|t)$ by back-iterating in time from the end of the trial. For most relevant tasks, to never report is always sub-optimal, so the value at $t = t_{\max}$ is set by the best of the two reporting ($\pm$) actions, which do not have a recursive dependence on the value and so can seed the recursion.

We now interpret this general formulation in terms of opportunity costs. For the choice of a static opportunity cost rate of time, $c_t = \rho$. This is the AR-RL case. As in [8], a constant auxiliary deliberation cost rate, $c$, incurred only up to decision time can be added, $c_t = \rho + c\Theta(t^{\mathrm{dec}} - t)$. Of course, $\rho$ is unknown *a priori*. For this solution method, its value can be found by exploiting the self-consistency constraint, $V(s|t = 0) = V(s|t = T + 1)$. This dependence can be seen formally by taking the action value Eq 20, choosing $a$ according to $\pi$ to obtain the state value, $V(s|t)$, and evaluating it for $t = 0$,

$$V(s|t = 0) \quad = \mathbb{E}_{t^{\mathrm{dec}}}\left[\sum_{t=0}^{T} R_t - \rho\right] + V(s|t = T + 1) \tag{22}$$

$$= \mathbb{E}_{t^{\mathrm{dec}}}\left[r(t^{\mathrm{dec}}) - \rho T(t^{\mathrm{dec}})\right] + V(s|t = T + 1) \tag{23}$$

$$= \bar{R} - \rho\bar{T} + V(s|t = T + 1) . \tag{24}$$

Here, $\bar{R} = \mathbb{E}_{t^{\mathrm{dec}}}[r(t^{\mathrm{dec}})]$ and $\bar{T} = \mathbb{E}_{t^{\mathrm{dec}}}[T(t^{\mathrm{dec}})]$ denotes the expectations over the trial ensemble that, when given the state sequence, transforms to an average over $t^{\mathrm{dec}}$, the trial decision time, defined as when $V(s|t)$ achieves its maximum on the state sequence, $(s_0, \ldots, s_{t_{\max}})$. The expected trial reward function, $r(t) := \max_{a \in \{-,+\}} r(s, a|t)$ is the expected trial reward for deciding at $t$. Imposing the self-consistency constraint on Eq 24 recovers the definition $\rho = \bar{R}/\bar{T}$.

## Asymmetric switching cost model

Here, we present the model component that accounts for the asymmetric relaxation timescales after context switches. The basic assumption is that tracking a signal at a higher temporal resolution should be more cognitively costly. A cost-aware strategy addressing this fact is to adapt the integration time to smaller values when resolution mismatch costs are higher so the system lowers this higher cost quickly. For the tokens task, adapting from faster to slower environments should then happen more quickly than the reverse. We now develop this idea formally (see S4 Fig).

Let $T_{\mathrm{track}}$ and $T_{\mathrm{sys}}$ be the timescale of tracking and of the tracked system, respectively. One way to interpret the mismatch ratio, $T_{\mathrm{sys}}/T_{\mathrm{track}}$, is via an attentional cost rate, $q$. This rate should decay with $T_{\mathrm{track}}$: the slower the timescale of tracking, the lower the cognitive cost. For simplicity, we set $q = 1/T_{\mathrm{track}}$ (S4(A) Fig). Integrating this cost rate over a characteristic time of the system is then the tracking cost, $Q = qT_{\mathrm{sys}} = T_{\mathrm{sys}}/T_{\mathrm{track}}$, which is also the mismatch ratio.

We propose that $Q$ enters the algorithm via a scale factor on the integration time of the reward filter for $\hat{\rho}_k^{\tau_{\text{context}}}$, $\tau_{\text{context}}$. We redefine $\tau_{\text{context}}$ as

$$\tau_{\text{context}} \leftarrow \frac{\tau_{\text{context}}}{1 + Q^v} \;, \tag{25}$$

where $v$ is a sensitivity parameter that captures the strength of the nonlinear sensitivity of the speed up (for $v > 1$) or slow down (for $v < 1$) in adaptation with the tracking cost, $Q$ (S4(A) Fig shows how this timescale varies over $Q$ for three values of $v$). We note that $v$ captures the meta-cognitive belief that the subject has about how cognitively effortful is its tracking behaviour. A natural choice for $T_{\text{sys}}$ is $T_k$, the trial duration. For $T_{\text{track}}$, we introduce the filtered estimate of the trial duration, $\hat{T}_k^{\tau_{\text{context}}}$ (computed using the same simple low-pass filter *c.f.* Eq 8). Thus, the tracking timescale adapts to the system timescale. As a result of how $\tau_{\text{context}}$ is lowered by $Q$ for $v > 1$, this adaptation is faster in the fast-to-slow transition relative to the slow-to-fast transition.

## Prediction for asymmetric rewards

Given a payoff matrix, $\boldsymbol{R} = (r_{s,a})$, where $r_{s,a}$ is the reward for reporting $a \in \{-, +\}$ in the trial realization leading to $s$, here the sign of $N_{t_{\max}}$, and the probability that the rightward choice is correct, $p_{n,t}^+$, the expected reward for the two reporting actions in a trial is given by the matrix equation

$$\begin{bmatrix} \langle r | a = +, n, t \rangle & \langle r | a = -, n, t \rangle \end{bmatrix} = \begin{bmatrix} p_{n,t}^+ & 1 - p_{n,t}^+ \end{bmatrix} \begin{bmatrix} r_{++} & r_{+-} \\ r_{-+} & r_{--} \end{bmatrix}.$$

Here, the corresponding reported choice is $a^* = \text{argmax}_{a \in \{-,+\}} \langle r | a, n, t \rangle$. In this paper and in all existing tokens tasks, $\boldsymbol{R}$ was the identity matrix. In this case, and for all cases where $\boldsymbol{R}$ is a symmetric matrix, $\boldsymbol{R} = \boldsymbol{R}^\top$, an equivalent decision rule is to decide based on the sign of $N_t$. When $\boldsymbol{R}$ is not symmetric, however, this is no longer a valid substitute. Asymmetry can be introduced through the actions and the states.

Using an additional parameter $\gamma$, we introduce asymmetry via a bias for $+$ actions that leaves the total reward unchanged by replacing the payoff matrix with

$$\boldsymbol{R}_{\text{asym}} = \begin{bmatrix} r_{++}(1 + \gamma) & r_{+-}(1 - \gamma) \\ r_{-+}(1 + \gamma) & r_{--}(1 - \gamma) \end{bmatrix},$$

The result for $\gamma = -0.6$, 0, and 0.6 is shown in S10 Fig. For $\gamma > 0$ the decision boundary for $a = +$ shifts up proportional to $\gamma$. For $\gamma < 0$ the decision boundary for $a = -$ shifts down proportional to $-\gamma$. The explanation is that the components are set and exchange where the decision is exchanged, $N_t = 0$ for the symmetric case. This changes to $N_t \propto \pm\gamma$ for the asymmetric $\gamma \neq 0$ case.

## Comparing reward rates and slopes of urgency

Reference [23] parametrize urgency with the saturation value, $u_\infty$, and the half-maximum, $\tau_{1/2}$. The initial slope is given by their ratio. We used the context-conditioned values published in Table 1 in [23] for the $n = 70$ (no 90° control) dataset. The context-conditioned reward rates, $\rho_\alpha$, are computed as the accuracy $\langle R \rangle_{|\alpha}$ divided by the average trial time, $\langle T \rangle_{|\alpha}$ for choice number $\alpha \in \{2, 4\}$ as context. We computed $\langle R \rangle_{|\alpha=2} = 0.71$ and $\langle R \rangle_{|\alpha=4} = 0.49$. The trial time is the sum of the response time, the added time penalty if incorrect, and the inter-trial interval. We computed the response times $t_{\text{response},\alpha=2} = 0.527$ and $t_{\text{response},\alpha=4} = 0.725$. While the dataset

contains the response times, it does not have the latter two. The time penalty was on the order of 1 second, as was the time penalty (A. Churchland. Personal communication). Under those estimates, the reward rates are $\rho_{\alpha=2} = 0.40$ and $\rho_{\alpha=4} = 0.22$. The ratio between slopes is 1.8 and the ratio of reward rates was 2.3 giving an error of about 20%.

## Supporting information

**S1 Video. Example simulation of PGD.**
(MP4)

**S1 Fig. Reward filtering scheme for online computation of within-trial opportunity cost.** With $t$ denoting absolute time, the reward sequence, $R_t$, is integrated on both a stationary ($\tau_{\text{long}}$) and context ($\tau_{\text{context}}$) filtering timescale to produce estimates of the stationary and context-specific reward rates, respectively. These are large and small, respectively, relative to the average context switching timescale, $T_{\text{block}}$. The estimate of the context-specific offset, $o_t$ is computed by time-integrating the difference of these two estimates. In this filtering, when a trial terminates, the effective operation is that $C_t^{\text{del}}$ is set to $o_t$, and the latter is zeroed. Thus, the opportunity cost starts at this offset and then integrates $\rho_{\text{long}}$, $C_{t,k}^{\text{del}} = o_{T_{k-1},k-1} + \rho_{\text{long},k-1}t$, where $o_{Tk-1}, k-1 = (\rho_{\text{context},k-1} - \rho_{\text{long},k-1})T_{k-1}$. Notes on the computational graph: Arrows pass the value at each time step (dashed arrows only pass the value when a trial terminates). Links annotated with '−' multiply the passed quantity by −1.
(PDF)

**S2 Fig. PGD agent plays the tokens task with periodic $\alpha$-dynamics.** (A) Trials are grouped into alternating trial blocks of constant $\alpha$ (fast (orange) and slow (blue) conditions). (B) Here, trial block durations are constant over the experiment. (C) Decision times over the trials from (A) distribute widely, but relax after context switches. (D) Block-averaged decision times remain stationary. Inset shows the context-conditioned trial-averaged reward $\langle R_k \rangle$ and trial duration $\langle T_k \rangle$ (orange and blue dots; black is unconditioned average; $\langle \cdot \rangle$ denotes the trial ensemble average). Lines pass through the origin (slope given by the respective reward rate). (E) Distribution of estimates have lower variance than the trial reward rates, $\rho^{\text{trial}}$ (gray). The conditioned averages of $\hat{\rho}_k^{\tau_{\text{context}}}$ shown as blue and orange. (F) The relative error in estimating $\rho$, $E_t = \frac{1}{t}\sum_k^t |\hat{\rho}_k^{\tau_{\text{long}}} - \rho|/\rho$, for $\tau_{\text{long}} = 10^3$(circle), $10^4$(square), $10^5$(triangle). Inset shows that $E_{T_{\text{exp}}} \propto (\tau_{\text{long}}/T_{\text{block}})^{-1}$ over a grid of $\tau_{\text{long}}$ and $T_{\text{block}}$ as expected (black line).
(PDF)

**S3 Fig. Comparison of PGD and NHP in non-stationary $\alpha$ dynamics from [25]: Subject 2.** Same as Fig 5.
(PDF)

**S4 Fig. Asymmetric switching cost model.** (A) Attentional cost rate, $q$, is set to be inversely proportional to tracking timescale, $T_{\text{track}}$. (B) Filtering timescale $\tau_{\text{context}}$ is scaled down with tracking cost, $Q = T_{\text{sys}}/T_{\text{track}}$ from a base timescale, here denoted $\tau_0$ (shown for three values of sensitivity $v = 2, 4, 8$).
(PDF)

**S5 Fig. Sigmoidal approximation to expected reward.** (A) the approximation explained in Methods: State-conditioned expected trial reward, for different decision times. (B) The error in the approximation for different decision times.
(PDF)

**S6 Fig. Model validation on behavioural statistics from [25].** (a,b) Running average (last 1000 trial) of trial reward rate $\rho_k^{\text{trial}}$. (c,d) Histograms of trial reward rate, $\rho_k^{\text{trial}}$ (C) and trial duration, $T_k$ (D). (E) Auto-correlation function of trial duration. (F) Data vs. model decision time (gray-scale is count; white dashed line is perfect correlation; actual Pearson correlation is shown).
(PDF)

**S7 Fig. Comparison of trial-aware and trial-unaware results.** (a,b) 1/2-Survival probability contours for subject 1 (dashed), trial-aware PGD (blue), and trial-unaware PGD (red) for slow (A) and fast (B) context-conditioned data. (C) Opportunity cost for trial-unaware PGD (compare with Fig 2B). Opportunity cost range adjusted here such that data within standard error of trial-unaware PGD model prediction for slow block (blue).
(PDF)

**S8 Fig. Comparison of PGD and AR-RL learning on a patch leaving task.** Performance is defined as relative regret rate, $(\hat{\rho} - \rho^*)/\rho^*$ (PGD (dots); AR-RL (lines)). (A) Performance over different sizes of the state vector ($d$ = 100 (blue), 200 (orange), 300 (green)). (B) Performance over different learning rates (parametrized by integration time constant, $\tau = 1 \times 10^4$ (blue), $2 \times 10^4$ (orange), $3 \times 10^4$ (green)). (parameters: $\lambda$ = 1/5; $r_{\text{max}}$ sampled uniformly on [0, 1]). A random state label permutation is made at the time indicated by the black arrow. Values were initialized at $-1$.
(PDF)

**S9 Fig. Reward rate optimal strategies in ($\alpha$, $c$) plane.** (A) The reward-rate maximizing policy interpolates from the wait-for-certainty strategy at weak incentive (low $\alpha$) and low deliberation cost (low $c$), to the one-and-done strategy at strong incentive (high $\alpha$) and high deliberation cost (high $c$). Dashed lines bound a transition regime between the two extreme strategies. Red line denotes where they have equal performance. (b-e) Slices of the ($\alpha$, $c$)-plane. Shown are the reward rate as a function of $\alpha$ (b,c) and $c$ (d,e) (wait-for-certainty strategy is shown in blue; one-and-done strategy is shown in orange). $N$ is the magnitude of the token difference.
(PDF)

**S10 Fig. Asymmetric action rewards skew survival probability.** Here, we plot the half-maximum of the PGD survival probability for three values of the action reward bias, $\gamma$ = −0.6, 0, 0.6 (blue, black and orange, respectively). Other model parameters same as in fitted model.
(PDF)

**S11 Fig. Decision times relative to token jumps.** Here, we plot the histograms of decision times using their position between token jumps, the step fraction. The data is separated by $\alpha$ and monkey.
(PDF)

## Acknowledgments

We would like to acknowledge helpful discussions with Jan Drugowitsch, Becket Ebitz, and Paul Masset, and to Anne Churchland for sharing data from [23].

## Author Contributions

**Conceptualization:** Maximilian Puelma Touzel, Guillaume Lajoie.

**Data curation:** Maximilian Puelma Touzel, Paul Cisek.

**Formal analysis:** Maximilian Puelma Touzel.

**Funding acquisition:** Guillaume Lajoie.

**Investigation:** Maximilian Puelma Touzel.

**Methodology:** Maximilian Puelma Touzel.

**Project administration:** Maximilian Puelma Touzel.

**Supervision:** Guillaume Lajoie.

**Writing – original draft:** Maximilian Puelma Touzel, Paul Cisek, Guillaume Lajoie.

**Writing – review & editing:** Maximilian Puelma Touzel, Paul Cisek, Guillaume Lajoie.

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
