## [Decision Letter · Decision Letter 0]

17 Dec 2021

Dear Dr. Puelma Touzel, 

Thank you very much for submitting your manuscript "Deliberation gated by opportunity cost adapts to context with urgency" for consideration at PLOS Computational Biology.

As with all papers reviewed by the journal, your manuscript was reviewed by members of the editorial board and by several independent reviewers. In light of the reviews (below this email), we would like to invite the resubmission of a significantly-revised version that takes into account the reviewers' comments.

We cannot make any decision about publication until we have seen the revised manuscript and your response to the reviewers' comments. Your revised manuscript is also likely to be sent to reviewers for further evaluation.

Sincerely,

Stefano Palminteri

Associate Editor

PLOS Computational Biology

Samuel Gershman

Deputy Editor

PLOS Computational Biology

Reviewer's Responses to Questions

**Comments to the Authors:**

Reviewer #1: This manuscript proposes a heuristic-based model to explain the speed-accuracy tradeoff in reinforcement learning (RL). The proposed model, Performance-Gated Deliberation (PGD), is an improvement over average-reward reinforcement learning (AR-RL) in that it can account for context-induced changes in the opportunity cost of time, whereas AR-RL assumes a fixed opportunity cost of time (i.e., the stationary average reward). Two types of opportunity cost are estimated according to PGD: the commitment and deliberation costs. While the former declines with increasing deliberation (due to the increasing probability of making a correct decision as evidence accumulates), the latter rises steadily (since longer deliberation lowers the amount of reward that can be obtained in a fixed time interval). Further, the rising deliberation cost is thought to be encoded in the basal ganglia as an urgency signal. PGD posits that a decision is made at the point in time when the decreasing commitment cost and increasing deliberation cost intersect. The model is supported with choice and neural data from two non-human primates performing the tokens task, and several analytical derivations are provided.

Overall, I thought the paper was well-written and provided a convincing case for the proposed model. I have only a few minor comments and suggestions.

1. If I understand correctly, deliberation times should be shorter in contexts with high average reward and longer in contexts with low average reward due to the opportunity cost of time. If this is the case, it would be helpful to state this explicitly somewhere in the Introduction so that readers who are not as familiar with the theory can better understand its implications.

2. Along those same lines, how do the results of the present study relate to the finding that deliberation times in RL are shorter in contexts with higher-valued options (e.g., Fontanesi et al., 2019)? Is this something that would be predicted by PGD?

3. Page 12, line 364: Please describe the wait-for-certainty and one-and-done strategies.

4. Figure 5 (B, C): Is it possible to test statistically whether the distribution of errors in slow and fast blocks differs significantly between PGD and AR-RL?

5. Figure 5 caption: For the survival probabilities, should it not be P(tdec > t | Nt, t) instead of P(tdec = t | Nt, t)?

6. Page 4, line 109: Can you clarify what you mean by an ensemble average? Is this just a weighted average?

References:

Fontanesi, L., Gluth, S., Spektor, M. S., & Rieskamp, J. (2019). A reinforcement learning diffusion decision model for value-based decisions. Psychonomic Bulletin and Review, 26, 1099-1121.

Reviewer #2: *Review uploaded as an attachment as well

The question of how organisms evaluate alternatives options relative to one another as well as with respect to shifting, context dependent benchmarks in order to simultaneously optimize choice and time is unquestionably one of the most central topics of research in computational cognitive science. The authors present an impressive and well thought out model which accomodates adaptive serial comparison inspired by groundbreaking modeling work using Performance Gated Deliberation to study simultaneous alternatives.

“These are significant, practical complications of making decisions contingent on opportunity costs, the formal economic concept capturing the value of the alternatives lost by committing a limited resource to a given use”. (line 23)

Within the logical flow of the introduction, which appears to follow the customary broad-to-specific funnelling of information, it seems as though the general concept of opportunity cost should be introduced prior to the more specific discussion of the opportunity cost of time.

It may also be useful to unpack this concept a bit further by providing a brief example in which the limited resource with alternative uses is not time (e.g., monetary currency, energy, memory/representational capacity, or attention as later sort of touched on around line 876 where the notion of the attentional cost rate q is introduced).

Indeed, as the authors hint at, every decision between mutually exclusive alternatives implies an opportunity cost, but in the case where alternatives are presented simultaneously, the limited resource is choice itself and the opportunity cost is the value of the next best alternative. Some further discussion or explanation of opportunity costs along this line seems necessary to prepare the reader for the subsequent introduction of the two, nested opportunity costs central to the PGD model.

“For example, animals will learn to value a given food resource differently depending on whether it is encountered during times of plenty versus scarcity.” (line 20)

While it is wise of the authors to give a nod towards the optimal foraging theory early on in the introduction, it may be imprudent not to address this literature more directly via explicit citation and description. In particular, rather than simply stating that valuation depends on the richness of the environment, consider a brief practical demonstration of how valuation depends on the richness of the environment (with emphasis on how this relationship arises from the scarcity of time).

To this end, an example of patch foraging (e.g., Krebs 1974) would be particularly decorous because the manner in which this framework operationalizes the allocation of scarce time is most analogous to optimization of deliberation time in your model (compared to an example prey selection such as Krebs et al., 1978).

“The agent's knowledge of and ability to track context thus influences the value it assigns to possible alternatives.”

Authors might add that factors that influence the perceived state value have been shown to have direct consequences on behavior.

The notion that the correspondence between optimal movement speed and opportunity cost might account for dopamine’s involvement in vigor has motivated recent work directly investigating whether a similar mechanism supports context-dependent valuation in computer-based analogues of patch foraging tasks. This work seems worth mentioning somewhere in the introduction.

Of particular relevance, people’s willingness to leave deplenished patches have been shown to be influenced by dopaminergic depletion and replacement in Parkinson’s disease (Constantino et al., 2017) and dopaminergic drugs in healthy participants (Le Heron et al., 2020).

variance of the contextual reward distribution

As the proposed model accounts not just for the mean, but also the variance of reward in the local context, it would be useful to cite evidence of the behavioral relevance of this variable (e.g., Bavard et al., 2018).

The foraging literature also provides evidence for the role of the variance of in the contextual reward distribution (e.g., Caraco et al., 1980).

“Performance-Gated Deliberation (PGD), that uses the increasing opportunity cost of time in a trial to collapse the decision boundary directly, by-passing the need to maximize relative value” (line 58)

This is only the latter part of a run-on sentence which begins in line 56, but in addition to breaking up this sentence into smaller, more palatable bites for the reader, the content of this latter part needs substantial clarification

“The strategies for subject 2 are qualitatively similar, but shifted to earlier times relative to subject 1. Our model explains this inter-individual difference as resulting from subject 2's larger reward bias and faster context integration” (line 394)

This is a very interesting point and should be further unpacked and spelled out for the reader.

Having explained the logic behind this interpretation more comprehensively, the authors may want to point back to this result directly in the discussion with respect to hypothetical advantages of potentially examining or parameterizing inter-individual differences in future work using this sort of model.

“The basic assumption is that tracking a signal at a higher temporal resolution should be more cognitively costly, so that adapting from faster to slower environments should happen more quickly than the reverse, so as to not pay this cost unnecessarily.” (line 871)

I think the wording is a bit misleading here. It seems like the authors are trying to say that the model accounts for asymmetric switching costs by assuming that the temporal resolution of tracking adapts to the statistics of the environment/context and due to this, behavior is more responsive to changes from fast to slow (as opposed to the reverse) because they occur when their is greater tracking resolution.

Out of curiosity, what do the authors propose the variable v to correspond to either cognitively or with respect to dopaminergic functioning or other mechanisms in the brain? Or rather, does v characterize something more closely related to the nature of the tracking problem which stems more innately from the task than any mechanisms in the brain.

References:

Bavard, S., Lebreton, M., Khamassi, M., Coricelli, G., & Palminteri, S. (2018). Reference-point centering and range-adaptation enhance human reinforcement learning at the cost of irrational preferences. Nature communications, 9(1), 1-12.

Caraco, T., Martindale, S., & Whittam, T. S. (1980). An empirical demonstration of risk-sensitive foraging preferences. Animal Behaviour, 28(3), 820-830.

Constantino, S. M., & Daw, N. D. (2015). Learning the opportunity cost of time in a patch-foraging task. Cognitive, Affective, & Behavioral Neuroscience, 15(4), 837-853.

Constantino, S. M., Dalrymple, J., Gilbert, R. W., Varanese, S., Di Rocco, A., & Daw, N. D. (2017). A neural mechanism for the opportunity cost of time. bioRxiv 173443.

Krebs, J. R. (1974). Colonial nesting and social feeding as strategies for exploiting food resources in the Great Blue Heron (Ardea herodias). Behaviour, 99-134.

Krebs, J. R., Kacelnik, A., & Taylor, P. (1978). Test of optimal sampling by foraging great tits. Nature, 275(5675), 27-31.

Le Heron, C., Kolling, N., Plant, O., Kienast, A., Janska, R., Ang, Y. S., ... & Apps, M. A. (2020). Dopamine modulates dynamic decision-making during foraging. Journal of Neuroscience, 40(27), 5273-5282.

Reviewer #3: The authors propose a new strategy (Performance-Gated Deliberation, PGD) that explains how animals decide "when to decide". They showed that this strategy outperforms the average-reward reinforcement learning (AR-RL) strategy to explain behavioral data of 2 non-human primates. In general, I think that this manuscript makes important contributions to the current literature, although a few things could be better clarified and I have some doubts over the generalizability of the results.

Major points:

A.

I found this manuscript very technical, to the point that it's hard to read (even considering the target of PLOS computational biology). For example, I would suggest at least explaining the token task earlier in the manuscript (perhaps by adding a figure, but also explain what are the contexts and what is the learning aspect of this task). Also, AR-RL might not be super obvious/familiar to most readers and it could be better introduced.

B.

The authors studied incentivized decisions where the time spent on the decision itself is a missed opportunity value, depending on the context. I wondered whether 1) it can be shown that subjects learn this across trials (with analyses or a figure); 2) there are any individual differences in this learning process that could be modeled?

C.

This brings me to my doubts about the chosen sample: Are these 2 monkeys representative of the behavior of other monkeys? And of humans as well? Figure 5A compares the 2 monkeys' behavior with a "reference expert human". Citation 31 also specifies "Single subject behavioural data shared by Thomas Thierry." This should be better explained. Is this published data? How representative are these data of human behavior?

D.

It is a bit unclear to me what tasks does PGD generalize at the moment. Even though this point is mentioned in the Discussion, I think it would make sense to fit it to more proper foraging data, or to a task where it can be shown how animals learn (and fail to learn) the optimal time to invest in a decision.

Minor points:

- The introduction paragraph confuses the idea of spending time doing something with the idea of spending time deciding (what to do or to choose). The examples they make are more in line with the former, while the topic of urgency has to do with the latter. I suggest changing the first paragraph.

- Line 147: "A foraging agent" not "An foraging agent"

- Figure 1d: I cannot distinguish the 2 colors

**Have the authors made all data and (if applicable) computational code underlying the findings in their manuscript fully available?**

Reviewer #1: **No: **There is a Github link provided for the simulation code but it appears to be broken.

Reviewer #2: Yes

Reviewer #3: **No: **Actaully, I am not sure whether the data are shared. There is a github link on the manuscript, but the link does not exist.

PLOS authors have the option to publish the peer review history of their article (what does this mean?). If published, this will include your full peer review and any attached files.

Reviewer #1: No

Reviewer #2: No

Reviewer #3: No
---

## [Decision Letter · Decision Letter 1]

5 Apr 2022

Dear Dr Puelma Touzel

We are pleased to inform you that your manuscript 'Performance-gated deliberation: A context-adapted strategy in which urgency is opportunity cost' has been provisionally accepted for publication in PLOS Computational Biology.

Best regards,

Stefano Palminteri

Associate Editor

PLOS Computational Biology

Samuel Gershman

Deputy Editor

PLOS Computational Biology

Reviewer's Responses to Questions

**Comments to the Authors:**

Reviewer #1: I commend the authors on addressing all of my concerns from the original manuscript. I have no further comments.

**Have the authors made all data and (if applicable) computational code underlying the findings in their manuscript fully available?**

Reviewer #1: None

PLOS authors have the option to publish the peer review history of their article (what does this mean?). If published, this will include your full peer review and any attached files.

Reviewer #1: No

---

## [Editor Report · Acceptance letter]

16 May 2022

PCOMPBIOL-D-21-01857R1 

Performance-gated deliberation: A context-adapted strategy in which urgency is opportunity cost

Dear Dr Puelma Touzel,

I am pleased to inform you that your manuscript has been formally accepted for publication in PLOS Computational Biology. Your manuscript is now with our production department and you will be notified of the publication date in due course.

With kind regards,

Olena Szabo
